# Radiocarbon analysis reveals underestimation of soil organic carbon persistence in new-generation soil models

Alexander S. Brunmayr[1], Frank Hagedorn[2], Margaux Moreno Duborgel[2,3], Luisa I. Minich[2,3], and Heather D. Graven[1]

[1]Imperial College London, Department of Physics, South Kensington Campus, London SW7 2AZ, England, United Kingdom
[2]Swiss Federal Institute for Forest, Snow and Landscape Research WSL, Zürcherstrasse 111, 8903 Birmensdorf, Zurich, Switzerland
[3]ETH Zurich, Department of Earth Sciences, Sonneggstrasse 5, 8092 Zurich, Zurich, Switzerland

**Correspondence:** Alexander S. Brunmayr (asb219@ic.ac.uk)

**Abstract.**

Reflecting recent advances in our understanding of soil organic carbon (SOC) turnover and persistence, a new generation of models increasingly makes the distinction between the more labile soil particulate organic matter (POM) and the more persistent mineral-associated organic matter (MAOM). Unlike the typically poorly defined conceptual pools of traditional SOC models, the POM and MAOM soil fractions can be directly measured for their carbon content and isotopic composition, allowing for fraction-specific data assimilation. However, the new-generation models' predictions of POM and MAOM dynamics have not yet been validated with fraction-specific carbon and $^{14}$C observations. In this study, we evaluate 5 influential and actively developed new-generation models (CORPSE, MEND, Millennial, MIMICS, SOMic) with fraction-specific and bulk soil $^{14}$C measurements of 77 mineral topsoil profiles in the International Soil Radiocarbon Database (ISRaD). We find that all 5 models consistently overestimate the $^{14}$C content ($\Delta^{14}$C) of POM by 69‰ on average, and 2 out of the 5 models also strongly overestimate the $\Delta^{14}$C of MAOM by more than 80‰ on average, indicating that the models generally overestimate the turnover rates of SOC and do not adequately represent the long-term stabilization of carbon in soils. These results call for more widespread usage of fraction-specific carbon and $^{14}$C measurements for parameter calibration, and may even suggest that some new-generation models might need to restructure or further subdivide their simulated carbon pools in order to accurately reproduce SOC dynamics.

## 1 Introduction

The terrestrial carbon reservoir sequesters an estimated 29% of anthropogenic $CO_2$ emissions each year (Friedlingstein et al., 2022), significantly reducing the accumulation rate of $CO_2$ in the atmosphere and thus slowing down climate change. However, the future role of the terrestrial carbon reservoir as a net $CO_2$ sink is uncertain, as Earth System Models (ESMs) produce a wide range of projections for the net land-atmosphere carbon flux over the course of the 21st century, partly due to high uncertainties in the carbon–climate feedback (Friedlingstein et al., 2014; Arora et al., 2020). Moreover, a study by He et al. (2016) using the radiocarbon ($^{14}$C) isotope suggests that some of the most widely used CMIP5 (Coupled Model Intercomparison Project

Phase 5) ESMs may be systematically overestimating the future land carbon sink, further casting doubt on the reliability of future land sink predictions. All five ESMs tested in their study strongly underestimated the $^{14}$C age of soil organic carbon, which indicates an overestimation of the simulated carbon cycling rates, particularly in the most stable soil carbon pools. After He et al. (2016) adjusted the soil carbon cycling rates to fit the observed $^{14}$C data, the ESMs ended up predicting $40 \pm 27\%$ lower carbon sequestration by the terrestrial sink in the 21st century than with their default parameters. This result puts into question the ability of current ESMs to accurately model soil carbon dynamics, and highlights the importance of validating model predictions with $^{14}$C data.

Almost all ESMs rely on soil organic carbon (SOC) modules that are ultimately based either on the Century model (Parton et al., 1987) (e.g., CESM2, Danabasoglu et al., 2020) or the RothC model (Coleman and Jenkinson, 1996) (e.g., JULES, Clark et al., 2011). Even though Century and RothC have been used for many decades to predict SOC dynamics in various landscapes with moderate success (Leifeld, 2008; Leifeld et al., 2008, 2009; Abramoff et al., 2022; Zhang et al., 2020), both modeling frameworks were developed in the 1980s, and thus reflect the comparatively limited understanding of soil carbon cycling of that time. Indeed, the model design of RothC is inspired by the now obsolete humification theory (Lehmann and Kleber, 2015; Schmidt et al., 2011), and neither RothC nor Century explicitly simulate specific processes of SOC cycling, such as physico-chemical protection of SOC or adsorption and desorption of dissolved organic carbon, because their mechanisms were previously not understood well enough.

According to our current understanding, the most important control on SOC stability is not so much the molecular composition or "quality" of organic matter, but rather its protection from microbial and abiotic decomposition through occlusion in aggregates and mineral association (Kleber et al., 2011; Dungait et al., 2012; Lehmann and Kleber, 2015; Lavallee et al., 2020). When SOC gets enclosed into aggregates or stabilized by interactions with reactive soil mineral surfaces of pedogenic oxides or phyllosilicates through cation bridging, electrostatic interactions, or the formation of inner- and outer-sphere complexes (Rasmussen et al., 2018a; Rowley et al., 2018; Vogel et al., 2014; Kleber et al., 2015), it becomes less accessible to decomposers and thus significantly increases its persistence in soils (Basile-Doelsch et al., 2020; Schrumpf et al., 2013; Doetterl et al., 2015). A new generation of SOC models is now being developed to incorporate the theory of SOC protection through occlusion and interactions with soil minerals into our carbon cycle predictions. A common feature of new-generation soil models is their distinction between particulate organic matter (POM) and mineral-associated organic matter (MAOM). The POM fraction largely consists of partially decomposed litter fragments smaller than $2\,\mathrm{mm}$ (Lavallee et al., 2020; Basile-Doelsch et al., 2020), which may be covered with a thin mineral coating (Wagai et al., 2009). On the other hand, the MAOM fraction contains organic matter chemically adsorbed onto reactive mineral surfaces, or stabilized by occlusion or adsorption inside micro-aggregates formed around sand, silt, or clay particles (Basile-Doelsch et al., 2020; Lavallee et al., 2020). Unlike the carbon pools of RothC and Century, the POM and MAOM fractions simulated by new-generation models are designed to be "measurable": they can be operationally defined with experimental protocols by which they can be separated from soil samples and then analyzed individually for their elemental and isotopic composition (von Lützow et al., 2007). This allows for a closer look into the processes governing soil carbon stabilization and for potentially much larger datasets for model calibration and validation. However, the

use of fraction-specific measurements to validate models is still limited, even for new-generation models (Zhang et al., 2021, Table S1).

The theory that protection and accessibility are the most important controls on SOC stability is strongly supported by [14]C studies (Gaudinski et al., 2000; Schrumpf et al., 2013, 2021), which could indicate that new-generation SOC models might perform better with [14]C than the traditional SOC models integrated into ESMs. [14]C is an effective carbon cycle tracer because it is chemically indistinguishable from the other carbon isotopes and therefore participates in the same carbon exchange mechanisms as the more abundant [12]C and [13]C isotopes. Over the past century, the atmospheric [14]C levels have undergone dramatic changes, most notably as a result of thermonuclear weapons tests in the 1950s and '60s, which have almost doubled the amount of atmospheric $^{14}CO_2$ in the Northern Hemisphere (see Figure 2). As this bomb-derived $^{14}CO_2$ spreads into the terrestrial carbon reservoirs through photosynthesis and into oceans through air-sea gas exchanges (Graven et al., 2020), the level of enrichment in bomb-derived [14]C across different terrestrial and oceanic carbon reservoirs helps to evaluate the speed and magnitude of carbon exchanges with the atmosphere on annual and decadal scales. Meanwhile for slower-cycling reservoirs such as deep soils or permafrost, the level of [14]C depletion due to radioactive decay (half-life of $5700 \pm 30$ years (Roberts and Southon, 2007)) helps to estimate the time scales of carbon stabilization in those reservoirs on the order of centuries and millennia. [14]C is therefore a powerful tool to study the exchanges and storage of carbon from decadal to millennial time scales. However, new-generation models do not generally implement [14]C simulations, and only a handful have systematically assimilated observed [14]C data (e.g., Tipping and Rowe, 2019; Braakhekke et al., 2014; Ahrens et al., 2020).

In this study, we use [14]C measurements of the organic carbon in the mineral topsoil to evaluate the performance of five new-generation SOC models: CORPSE (Sulman et al., 2014), MEND-new (Wang et al., 2022), Millennial v2 (Abramoff et al., 2022), MIMICS-CN v1.0 (Kyker-Snowman et al., 2020), and SOMic 1.0 (Woolf and Lehmann, 2019). These models were chosen because they are open source, actively developed, and influential in the soil modeling community. Leveraging the measurability of their pools, we compare these models' predictions to [14]C measurements of POM and MAOM, in addition to the total soil [14]C. This provides a detailed picture of the modeled SOC dynamics and enables us to carry out an in-depth analysis of the models' performances.

## 2 Methods

Throughout this paper, we report the [14]C content of a given carbon sample as $\Delta^{14}C$, which is the deviation of the sample's $^{14}C/^{12}C$ ratio from the "modern" standard, corresponding to the pre-industrial atmospheric $^{14}CO_2/^{12}CO_2$ ratio (Trumbore et al., 2016).

### 2.1 Fraction-specific carbon and radiocarbon measurements

We compare model predictions to three types of measured data for the topsoil: (1) the total SOC stocks in the topsoil, (2) the relative mass contributions of POM and MAOM to the total SOC stocks, and (3) the $\Delta^{14}C$ of POM, MAOM, and bulk SOC.

For this study, we use the International Soil Radiocarbon Database (ISRaD) (Lawrence et al., 2020) for carbon and $^{14}$C measurements of POM and MAOM obtained from soil samples using a combination of density fractionation and ultra-sonication. Density fractionation with ultra-sonication is currently one of the most effective and commonly employed methods for isolating POM and MAOM (Golchin et al., 1994; Griepentrog et al., 2015, 2014; Cerli et al., 2012; von Lützow et al., 2007; Poeplau et al., 2018). This method separates the soil into three "density fractions" – the free light fraction, occluded light fraction, and heavy fraction – in a three step process: (1) obtain the free light fraction from the soil sample by density fractionation; (2) in the remaining sample, destroy loosely-bound aggregates with ultra-sonication, thus releasing the occluded fraction; (3) isolate the occluded light fraction from the relatively denser heavy fraction by density fractionation. The resulting free and occluded light fractions, jointly referred to as the light fraction, correspond approximately to the POM, while the heavy fraction is a good proxy for MAOM (Mikutta et al., 2019; Lavallee et al., 2020).

ISRaD provides carbon and $^{14}$C data for the bulk soil, and the free light, occluded light, and heavy fractions. We directly associate MAOM with the heavy fraction in ISRaD, and POM with the light fraction (i.e., the sum of the free and occluded light fractions in ISRaD, see Appendix A1). When the $\Delta^{14}$C of the bulk soil is not measured or reported in ISRaD, we calculate it with a weighted average of the light and heavy fractions' $\Delta^{14}$C (see Appendix A2). In this study, we evaluate models only for the topsoil, which we strictly define as at least the top $5\,\text{cm}$ and at most the top $10\,\text{cm}$ of the mineral soil (see Appendix A3 for more details). This way, we can ignore the effect of vertical mixing of soil carbon, which plays a more important role in deeper soil $^{14}$C dynamics (Koven et al., 2013; Chen et al., 2019; Braakhekke et al., 2011, 2014), and instead focus more on the effectiveness of the model designs in terms of their simulated carbon pools and biochemical processes. Furthermore, by choosing such a narrow depth interval, we can treat the topsoil as one single homogeneous soil layer, which allows us to also evaluate models which are not vertically resolved and are only intended for topsoil simulations. The current version of ISRaD (v 2.5.5.2023-09-20, International Soil Radiocarbon Database, 2023) contains complete $^{14}$C datasets of the light and heavy density fractions in the topsoil of 77 soil profiles spread across 39 sampling sites, covering forests, shrubland, cultivated landscapes, and rangeland and grassland. See Appendix A3 for more information on the choice of profiles, and Appendix A4 for the derivation of the topsoil carbon and $^{14}$C data from layer data. Almost all of the sampling sites are in North America and Europe, and the remaining sites are located in Hawaii and Puerto Rico (see map in Figure 1). The dataset does not contain any permafrost, thermokarst, peatland, or wetland soils, and 75 of the 77 samples are from 1997–2015, with only one sample from 1949 and one sample from 1978. As shown in Figure 2, most datapoints bear a positive $\Delta^{14}$C value, demonstrating an enrichment in bomb-derived $^{14}$C in the topsoil. See Table S.4 in the Supplementary Material for more details on the data and the data sources for the 77 selected soil profiles.

## 2.2 Selection of new-generation models

We reviewed the literature to find new-generation models whose pools are fully compatible with the POM–MAOM distinction, and that are capable of running global simulations (i.e., their parameter values depend on the environmental conditions and are not just optimized for a few specific sites). Table 1 gives an overview of the features and capabilities of such new-generation models, almost all of which have been developed starting in the 2010s. Many new-generation SOC models also explicitly

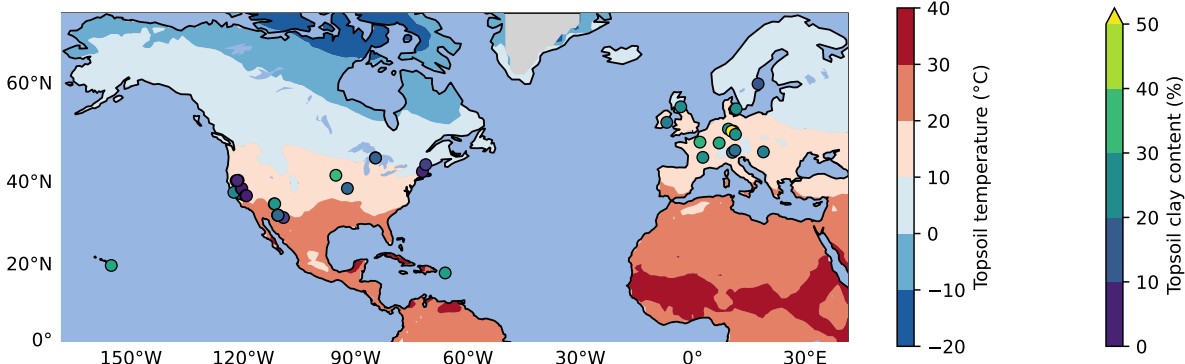

**Figure 1.** Map of selected topsoil sampling sites from ISRaD (Lawrence et al., 2020). 37 of the 39 sites are located in North America and Europe, and the two remaining sites are in Hawaii and Puerto Rico. All sites have a complete $^{14}$C dataset for the bulk soil and density fractions in the top 5 or 10 cm of the mineral soil. The map also shows two of the most important environmental controls on soil carbon persistence: soil temperature (at 4 cm depth, averaged over 1970–2010 period, 1 degree horizontal resolution) from the CESM2 Large Ensemble product (Rodgers et al., 2021) on the map background, and clay content in the topsoil from ISRaD or SoilGrids (Poggio et al., 2021) for each sampling site.

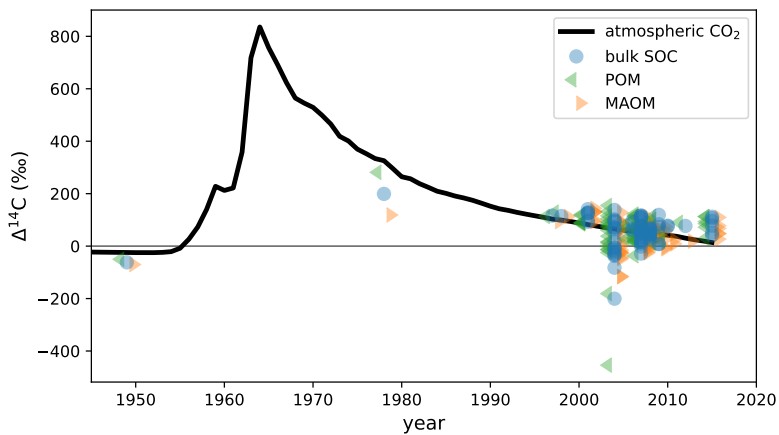

**Figure 2.** Measured $\Delta^{14}$C data of the POM and MAOM density fractions and total soil organic carbon (bulk SOC) at the selected topsoil profiles from ISRaD (Lawrence et al., 2020), overlaid on the atmospheric $\Delta^{14}$CO$_2$ curve of the Northern Hemisphere (Graven et al., 2017). All POM and MAOM fractions shown here were produced using the method of density fractionation with ultra-sonication. These ISRaD data were originally published in Baisden et al. (2002); Berhe et al. (2012); Harden et al. (2002); Heckman (2010); Heckman et al. (2018); Lybrand et al. (2017); Marín-Spiotta et al. (2008); McFarlane et al. (2013); Meyer et al. (2012); Rasmussen et al. (2018b); Schrumpf et al. (2013).

represent the microbial biomass as a separate carbon pool, since microbes are the main drivers of SOC turnover (Crowther et al., 2019; Basile-Doelsch et al., 2020; Schimel, 2023). The newest version of the MEND model simulates a variety of microbial exo-enzyme pools in addition to its microbial biomass pools (Wang et al., 2022). About half of the models listed in Table 1 already implement $^{14}$C simulations. However, none of them have systematically assimilated fraction-specific $^{14}$C data, instead relying on $^{14}$C data of bulk SOC or $^{14}CO_2$ data from soil respiration.

For this $^{14}$C study, we chose to evaluate the following models, as they are open-source and still actively developed:

- Millennial v2 (with Michaelis-Menten kinetics), Abramoff et al. (2022),

- SOMic 1.0, Woolf and Lehmann (2019),

- MEND-new (with default equations), Wang et al. (2022),

- CORPSE-fire-response (as implemented in Sulman, 2024a), Sulman et al. (2014),

- MIMICS-CN v1.0, Kyker-Snowman et al. (2020).

Figure 3 shows the general structure of the above models. All the selected models have pools which can be associated to the POM and MAOM fractions (see Appendix C for details on how we associate the pools to each fraction), and they all have at least one microbial biomass pool. We generally chose to evaluate the most recent version of each model. However, we found an error in the $^{14}$C implementation of the most recent version of MIMICS (Wang et al., 2021) (see Appendix E2), so we chose to use the coupled carbon-nitrogen version MIMICS-CN published one year prior in Kyker-Snowman et al. (2020). See Appendix B and Figures C1–C5 for more details on the exact versions and implementations of each model. Appendix D explains how we re-implemented the models to produce $^{14}$C predictions.

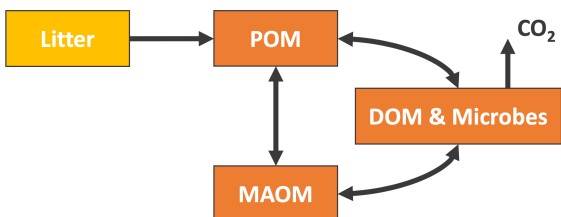

**Figure 3.** General structure of the new-generation models which we chose for this study. The MIMICS and CORPSE models additionally feature a $CO_2$ flux leaving MAOM and POM, which depends on the carbon use efficiency of the microbes. The SOMic and CORPSE models do not allow any flux from the DOM, Microbes, or MAOM back into the POM. More detailed diagrams for the MEND, Millennial, SOMic, CORPSE, and MIMICS models are shown in Figures C1–C5. Abbreviations: POM, particulate organic matter; MAOM, mineral-associated organic matter; DOM, dissolved organic matter.

**Table 1.** Summary of features and capabilities of new-generation models. All of the listed models are compatible with the distinction between POM and MAOM and are capable of running global simulations. The models selected for evaluation with [14]C in this study are indicated with an asterisk (∗). The first two columns are the year of the first publication and, if applicable, the year of the latest published revision of each model at the time of writing. The "Open-source", "Implements [14]C", and "Explicitly models" columns are checkmarked if at least one version of the model has open-source code, implements [14]C simulations, or explicitly models a specified pool or feature, respectively. In the "Vertical mixing" subcolumn, models with a downward arrow (↓) simulate any kind of downward transport or leaching for at least one of their pools, often in dissolved form, and sometimes using an advection equation. Models featuring an up–down arrow (↕) additionally implement vertical mixing with a diffusion equation for at least one of their pools.

| Model name | First publication | Latest revision | Open-source | Implements [14]C | Explicitly models | | | | Notes |
|---|---|---|---|---|---|---|---|---|---|
| | | | | | DOM | Microbes | Enzymes | Vertical mixing | |
| ∗ Millennial [1] | 2018 | 2022 | ✓ | | ✓ | ✓ | | ↓ | |
| ∗ SOMic [2] | 2019 | | ✓ | ✓ | ✓ | ✓ | | ↓ | |
| ∗ MEND [3] | 2013 | 2022 | ✓ | ✓ | ✓ | ✓ | ✓ | | [14]C only in 2015 |
| ∗ CORPSE [4] | 2014 | 2020 | ✓ | | | ✓ | | | |
| ∗ MIMICS [5] | 2014 | 2021 | ✓ | ✓ | | ✓ | | ↓↕ | [14]C and ↓↕ only in 2021 |
| MIND [6] | 2021 | | ✓ | | | ✓ | | | only a subset can be run globally[†] |
| AggModel [7] | 2013 | | ✓ | | | | | | incubation model |
| JSM [8] | 2020 | | (✓) | ✓ | ✓ | ✓ | | ↓↕ | source code accessible upon request |
| COMISSION [9] | 2015 | 2020 | | ✓ | ✓ | ✓ | | ↓↕ | [14]C introduced in v2.0 |
| Tipping & Rowe [10] | 2019 | | | ✓ | ✓ | | | ↓ | |
| MEMS [11] | 2019 | 2021 | | | ✓ | ✓ | | ↓↕ | ↕ introduced in v2.0 |
| SOMPROF [12] | 2011 | 2014 | | ✓ | | | | ↓↕ | [14]C introduced in 2014 |
| CAST [13] | 2013 | | | | | | | ↓ | |
| Struc-C [14] | 2009 | | | | | | | | |
| PROCAAS [15] | 2020 | | | | | | | | incubation model |

[1]Abramoff et al. (2018, 2022) | [2]Woolf and Lehmann (2019) | [3]Wang et al. (2013, 2015, 2022) | [4]Sulman et al. (2014, 2017); Salazar et al. (2018); Hicks Pries et al. (2018); Moore et al. (2020) | [5]Wieder et al. (2014, 2015); Zhang et al. (2020); Kyker-Snowman et al. (2020); Wang et al. (2021) | [6]Fan et al. (2021) | [†]Only the microbial necromass pools of MIND were run globally; some of the parameters (e.g., $V_{\mathrm{max},P}$ and $K_{M,P}$) necessary to run the live microbial biomass and plant-derived carbon pools do not have fitted values outside of 4 experimental test cases. | [7]Segoli et al. (2013) | [8]Yu et al. (2020) | [9]Ahrens et al. (2015, 2020) | [10]Tipping and Rowe (2019) | [11]Robertson et al. (2019); Zhang et al. (2021) | [12]Braakhekke et al. (2011, 2013, 2014) | [13]Stamati et al. (2013) | [14]Malamoud et al. (2009) | [15]Liu et al. (2020)

## 2.3 Model input data

For each measurement site, the models are run for the topsoil with local environmental forcing data from 1850 to 2014. The initial conditions in 1850 are found by spinning up the models, looping over a "pre-industrial" year, where the forcing data is averaged over the 1850–1879 period, until the system reaches equilibrium, i.e. does not experience any significant inter-annual variability. In practice, the carbon-nitrogen component of the MEND model is spun up from its default initial condition for 400 years and its $^{14}C$ component for 1000 years, the SOMic model is spun up for 50,000 years, and the remaining three models are spun up for 200 years from their pre-industrial steady-state solution. More details on the spinup methods for each model are given in Appendix B.

The selected models require a number of constant and time-dependent forcing data to be run at each study site. We assume that soil properties such as sand, clay and silt content, soil density, and land use are time-invariant since pre-industrial times. Where these site-specific soil properties are not reported in ISRaD, they are taken from the SoilGrids database (Poggio et al., 2021), accessed with the `soilgrids` python package, v0.1.4 (Gan, 2023). The MIMICS model also requires the lignin content of litter inputs, which we set to be a constant value depending only on the land use type. We assume that the lignin content is 25% for forest litter and 7% for shrubland litter (Rahman et al., 2013, Table 1). For grassland and cultivated landscapes, we assume a lignin content of 9% based on measurements of grasses at the seeding stage (Armstrong et al., 1950, Table 1). Weather-dependent and other dynamic environmental properties, such as soil temperature and $^{14}C$ influx, are taken from global model predictions with monthly time resolution. We use the monthly averaged CESM2 Large Ensemble (CESM2-LE) product (IBS Center for Climate Physics et al., 2021; Rodgers et al., 2021) for vertically resolved soil temperature and moisture, above- and below-ground net primary production (NPP), total gross primary productivity (GPP), litterfall and litter heterotrophic respiration, and the carbon-to-nitrogen ratio and $\Delta^{14}C$ of total litter carbon from 1850 to 2014 with 1 degree spatial resolution. Since the below-ground NPP from the CESM2-LE output is not vertically resolved, we derive the topsoil portion of the below-ground NPP using the exponential function model from Xiao et al. (2023). For nitrogen deposition rates, we use monthly data simulated by the NCAR Chemistry-Climate Model Initiative (CCMI) on a 0.5 degree grid from 1860 to 2016 (Tian et al., 2018) downloaded from the ISIMIP Repository (ISIMIP; Rosenzweig et al., 2017). We extend these data back to 1850 by setting the monthly nitrogen deposition rates for the 1850–1860 period to be equal to the average monthly rates over the 1860–1870 period.

Since none of the selected soil models represent lateral carbon transport or upward vertical mixing of soil carbon, the simulated topsoil systems receive all of their carbon exclusively from vegetation inputs. We can therefore estimate the $\Delta^{14}C$ of the carbon influx into the soil with the $\Delta^{14}C$ of litter from the CESM2-LE product. These litter $\Delta^{14}C$ data account for the pre-aging of carbon in vegetation (Herrera-Ramírez et al., 2020; Solly et al., 2018) because the litter carbon first passes through the vegetation pools in the land module of CESM2 (CLM5, Lawrence et al., 2019). For Millennial, CORPSE, and MIMICS, we estimate the carbon influx into the soil with the topsoil NPP, setting the slightly negative NPP values in the CESM2-LE output to zero. In the case of the MEND model, we use total GPP instead of NPP as a model input, as prescribed by MEND's developers (Wang, 2024). SOMic is the only model to require the use of litter inputs instead of NPP or GPP as a model input.

Following the example of the global simulations performed in SOMic's original publication (Woolf and Lehmann, 2019), we estimate litter inputs as the annual average of litterfall minus litter heterotrophic respiration, setting litter inputs to zero in the rare instances where annual litterfall is less than annual litter heterotrophic respiration. We derive the topsoil portion of litter inputs assuming they have the same vertical distribution as NPP.

## 3  Results

We produced carbon and $^{14}$C predictions with the MEND, Millennial, SOMic, CORPSE and MIMICS models for the 77 selected soil profiles, and compared them to the observed carbon and $^{14}$C data from ISRaD. Our main performance metrics are the root mean squared error (RMSE) and mean bias of the predictions with respect to the observational datasets described in Section 2.1. Table 2 gives a summary of the model performances. Detailed tables of the results, and plots of predictions against observations for each variable and each model can be found in the Supplementary Material (Tables S.3 and S.5, and Figures S.3). Note that the MEND model failed to run on 9 of the 77 selected soil profiles due to some numerical instability, and was unable to produce $^{14}$C data for 6 other profiles. Note also that the SOC stocks are not available for 17 of the 77 selected profiles.

**Table 2.** Root mean squared error (RMSE) and mean bias for each model with respect to each dataset. In the case of the MEND model, the RMSE and bias were calculated based on results of $n = 62$ profiles for the $\Delta^{14}$C rows, $n = 55$ for the SOC stocks, and $n = 68$ for the rows of POM and MAOM contributions. For all other models, $n = 77$ for all rows, except SOC stocks, where $n = 60$.

| | | MEND | Millennial | SOMic | CORPSE | MIMICS | Average |
|---|---|---|---|---|---|---|---|
| Bulk SOC $\Delta^{14}$C (‰) | RMSE | 84 | 115 | 122 | 77 | 80 | 96 |
| | Bias | +59 | +69 | +13 | +43 | 0 | +37 |
| POM $\Delta^{14}$C (‰) | RMSE | 94 | 119 | 105 | 119 | 129 | 113 |
| | Bias | +50 | +63 | +64 | +87 | +80 | +69 |
| MAOM $\Delta^{14}$C (‰) | RMSE | 103 | 117 | 116 | 61 | 74 | 94 |
| | Bias | +83 | +81 | +18 | +7 | −39 | +30 |
| SOC stocks (kgC m$^{-2}$) | RMSE | 4.0 | 3.8 | 1.9 | 6.5 | 2.3 | 3.7 |
| | Bias | −1.3 | +2.7 | +0.3 | +4.1 | −1.6 | +0.9 |
| POM contribution (%) | RMSE | 35 | 40 | 34 | 24 | 17 | 30 |
| | Bias | +22 | −33 | −26 | +12 | −2 | −5 |
| MAOM contribution (%) | RMSE | 34 | 41 | 33 | 22 | 21 | 30 |
| | Bias | −22 | +35 | +25 | −10 | −9 | +4 |

### 3.1  Carbon stocks and partitioning between pools

While the Millennial and CORPSE models tend to overestimate the topsoil SOC stocks of the selected soil profiles, MEND and MIMICS generally underestimate the SOC stocks (see Figure 4a). The SOMic model, which is the only model to estimate

carbon inputs into soils with litter inputs instead of primary productivity, produces the best predictions for the topsoil SOC

stocks with a positive mean bias of only $0.3\,\mathrm{kgC\,m^{-2}}$ (+13% relative to the observational mean) and a RMSE of $1.9\,\mathrm{kgC\,m^{-2}}$.

With the exception of the MIMICS model, the new-generation models generally fail to simulate the full range of variability in the observations of SOC partitioning between POM and MAOM (Figure 4b–c). The Millennial model's partitioning is nearly fixed around 8% POM and 92% MAOM for all sites, never deviating more than 1.5 percentage points from those values. The CORPSE and MIMICS models produce the best predictions of POM and MAOM contributions to the total SOC stocks. They

follow the one-to-one line of model predictions versus observations much better than the other models (see Figures S.3.2 and S.3.4 in the Supplementary Material), and they both have a RMSE around 20 percentage points and a bias of around 10 points or less in magnitude (Table 2). In comparison, the MEND, Millennial, and SOMic models have an average RMSE of 36 points and an average absolute bias of 27 points in their predictions of POM and MAOM contributions (Table 2).

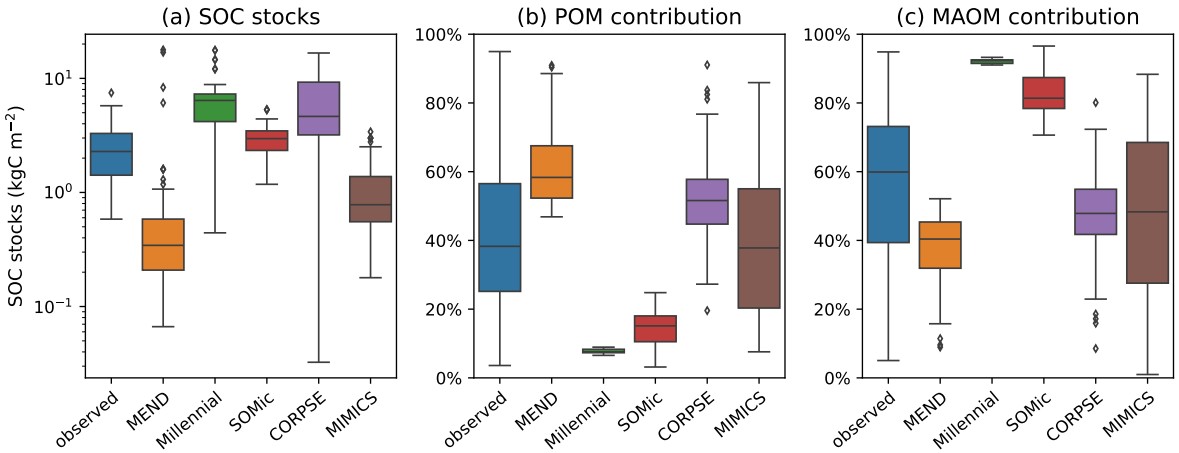

**Figure 4.** Observed and modeled SOC stocks in the topsoil (top 5 or 10 cm of mineral soil) plotted on a log-transformed axis in subplot (a), and contributions of the POM and MAOM fractions to the topsoil SOC stocks in subplots (b) and (c), respectively. Black diamonds are outliers. In subplot (a), $n = 60$ for the boxplot of observed data, $n = 68$ for MEND, and $n = 77$ for all other models. In subplots (b) and (c), $n = 77$ for all boxplots, except for MEND, where $n = 68$.

## 3.2   Performance with $^{14}$C

With the notable exception of MIMICS, the new-generation models consistently overestimate the $\Delta^{14}$C of bulk SOC, and their $^{14}$C predictions do not capture the full variability of the observations (see Figure 5a). This is reminiscent of the ESMs' $^{14}$C predictions from He et al. (2016), which also overestimate the $\Delta^{14}$C of SOC and underestimate its variability, though to a different extent and over a larger depth interval (top 1 m instead of the top 5 or 10 cm of the mineral soil). Our results could therefore suggest that the new generation of soil models may be facing similar issues as the traditional SOC models

incorporated into ESMs.

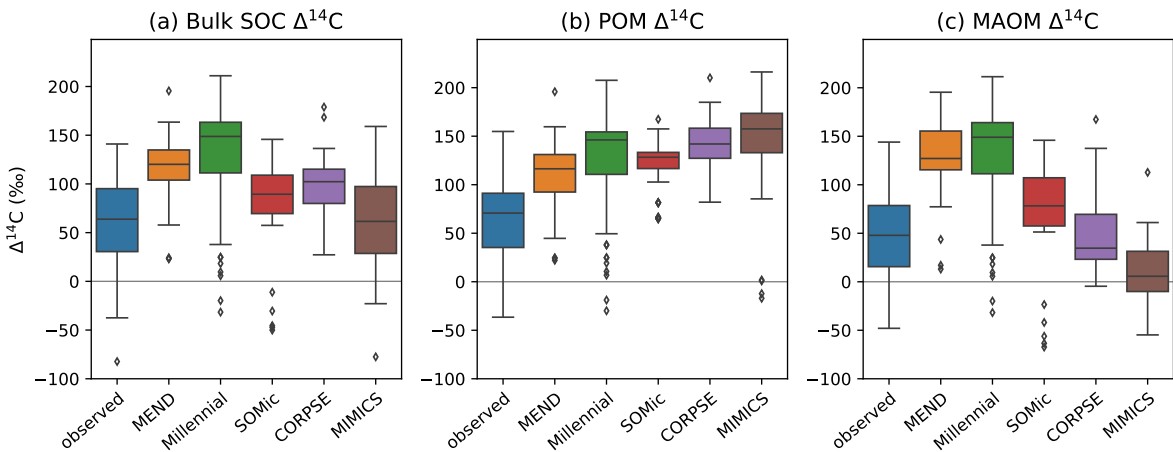

**Figure 5.** Observed and modeled $\Delta^{14}C$ of bulk SOC (a), POM (b), and MAOM (c) in the topsoil (top 5 or 10 cm of mineral soil). Black diamonds are outliers. Note that some extreme outliers are outside of the plotting range. To have a uniform and consistent $^{14}C$ dataset, we excluded the 1949 and 1978 samples so that we end up with more compact data spanning only 18 years at the tail end of the bomb spike. Therefore, $n = 75$ for all boxplots, except for MEND's, where $n = 60$.

The pool-specific $^{14}C$ results, shown in Figure 5b-c, shed a more critical light on the performance of MIMICS with the $\Delta^{14}C$ of bulk SOC. MIMICS overestimates the $\Delta^{14}C$ of POM by 80‰ and underestimates the $\Delta^{14}C$ of MAOM by around 40‰ on average, and these biases happen to cancel out in such a way that MIMICS produces very good predictions for the $\Delta^{14}C$ of bulk SOC with a RMSE of just 80‰ and no bias, the best performance among the evaluated models (see Table 2). All five models overestimate the $\Delta^{14}C$ of POM, with an average positive bias of 69‰, and MEND and Millennial also strongly overestimate the $\Delta^{14}C$ of MAOM by more than 80‰. CORPSE is good at predicting the $\Delta^{14}C$ of MAOM with effectively no bias, but its POM $\Delta^{14}C$ predictions have the largest bias (+87‰) among all the models. On average, the evaluated models have a positive bias between 37‰ and 69‰, and a RMSE around 100‰ in their $\Delta^{14}C$ predictions for POM, MAOM, and bulk SOC (see Table 2 for more details).

The models produce contrasting predictions for the evolution of soil $^{14}C$ over the second half of the 20th century. In Figure 6, we can see in a representative example of the model biases that the CORPSE, SOMic and MIMICS models produce very distinct $^{14}C$ dynamics for POM and MAOM, with POM having a predicted $\Delta^{14}C$ at least 200‰ higher than MAOM in the 1980s. On the other hand, the $\Delta^{14}C$ curves of MAOM and POM predicted by the MEND and Millennial models remain very close to each other throughout the post-bomb period. This is because Millennial and MEND have faster turnover rates than the other models, and their pools rapidly exchange carbon between themselves, thus homogenizing the $^{14}C$ signal across their simulated soil fractions (see Appendix F for more details on the turnover rates in Millennial, which are particularly fast).

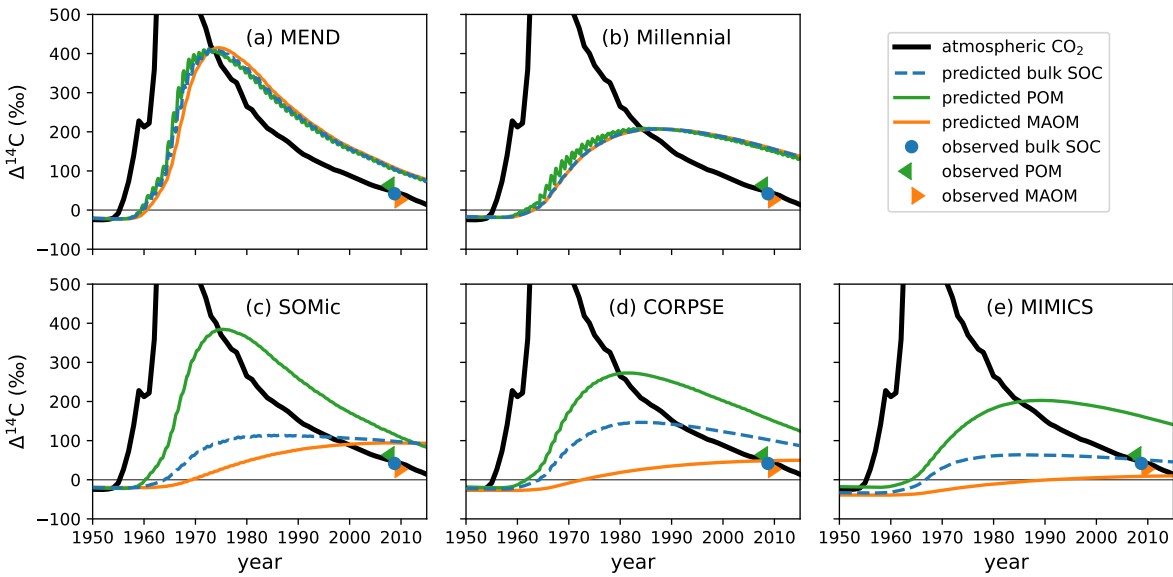

**Figure 6.** Observed and predicted $\Delta^{14}C$ of POM, MAOM, and bulk SOC in the top 10 cm of the mineral soil of an abandoned alpine grassland in the Stubai valley, Austria. The observed $^{14}C$ data from 2008 are published in Meyer et al. (2012), where the "observed POM" and "observed MAOM" data come from light and heavy density fraction measurements, respectively. The atmospheric $\Delta^{14}CO_2$ of the Northern Hemisphere (Graven et al., 2017) is shown for reference. With SOMic, CORPSE and MIMICS, the predicted $\Delta^{14}C$ of POM is distinct from the predicted $\Delta^{14}C$ of MAOM. On the other hand, the POM and MAOM fractions in MEND and Millennial have very similar $\Delta^{14}C$ throughout the bomb-spike period. Plots of the predicted and observed $\Delta^{14}C$ of all the other profiles are provided in the Supplementary Material (Figures S.2).

## 3.3 Role of environmental parameters

We further investigate how simulations depend on soil temperature and clay content, as these are considered some of the most important factors controlling SOC turnover and persistence (Basile-Doelsch et al., 2020; Leifeld et al., 2009).

Higher soil temperatures enhance microbial activity and generally increase the turnover rate of carbon in soils (German et al., 2012; Leifeld et al., 2009; Sierra et al., 2015). While the observed SOC stocks and POM and MAOM contributions are not correlated with temperature (Figure 7a–c), the observed $\Delta^{14}C$ of POM, MAOM, and bulk SOC significantly increase with higher temperature (Figure 7d–f). In contrast, the predicted $\Delta^{14}C$ of POM, MAOM, and bulk SOC are either uncorrelated or negatively correlated with soil temperature. All of the selected models modify carbon decomposition rates with a temperature-

dependent scaling factor (Abramoff et al., 2022; Woolf and Lehmann, 2019; Kyker-Snowman et al., 2020; Wang et al., 2022; Sulman et al., 2014), but these results could indicate that they may need to increase or change the effect of temperature on carbon turnover rates.

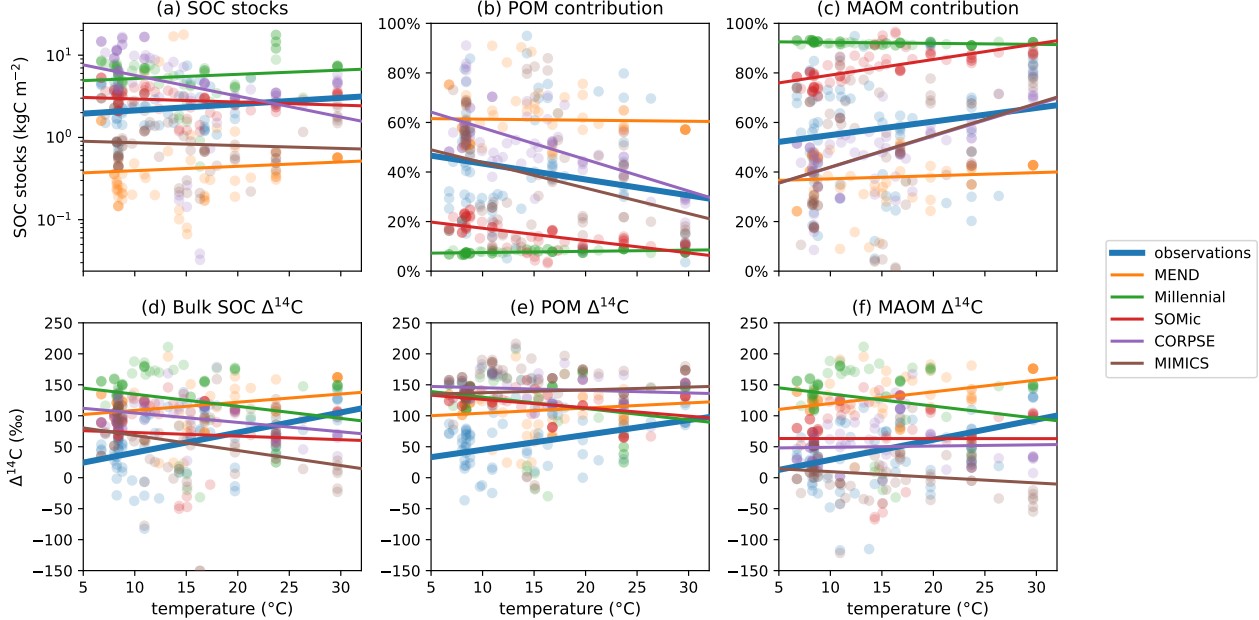

**Figure 7.** Relationships of observed and predicted carbon and $\Delta^{14}$C data with respect to mean annual temperature of the topsoil (averaged over the 1970–2010 period). Circles are datapoints, and lines are best linear fits through the points. The observed $\Delta^{14}$C of bulk SOC, POM, and MAOM have a strong positive relationship with temperature. Meanwhile, the predicted $\Delta^{14}$C are more weakly and sometimes negatively correlated with temperature. The linear fit line of CORPSE in subplot (c) is completely covered by the linear fit line of MIMICS. Note that some extreme outliers are outside of the plotting range, and that we once again excluded the 1949 and 1978 samples for these plots. Separate plots for each individual model are provided in the Supplementary Material (Figures S.1.2.32–36).

In Figure 8c, the clay content of the sampled topsoils seems to be a decisive factor controlling the observed contribution of MAOM carbon to the SOC stocks, with higher clay content correlating with higher MAOM contribution. This is also true
for the modeled MAOM contributions predicted by the MIMICS and CORPSE models, which produce the most accurate predictions of MAOM contribution (see Table 2). However, MIMICS appears to struggle with correctly simulating the effects of increased clay content on overall SOC dynamics, as evidenced by the inaccurate relationships of SOC stocks and $\Delta^{14}$C with clay (see Figure 8a and Figure 8d–f). It appears that MIMICS correctly reproduces the evolution of MAOM contribution with clay content by increasing the turnover time of carbon in MAOM, which in turn lowers the $\Delta^{14}$C of MAOM and increases
SOC stocks, contrary to the observations.

It is important to note that the regression lines in the $\Delta^{14}$C plots in Figures 7d–f and 8d–f could potentially be biased due to the different sampling years of soil profiles with different environmental parameters. However, those biases most likely do not affect our analysis of the results (see Appendix G, and plots with "normalized" $\Delta^{14}$C data in Figures S.1.1). The Supplementary Material also contains figures of the model biases and absolute errors plotted against temperature and clay content (Figures
S.1.2.1–24).

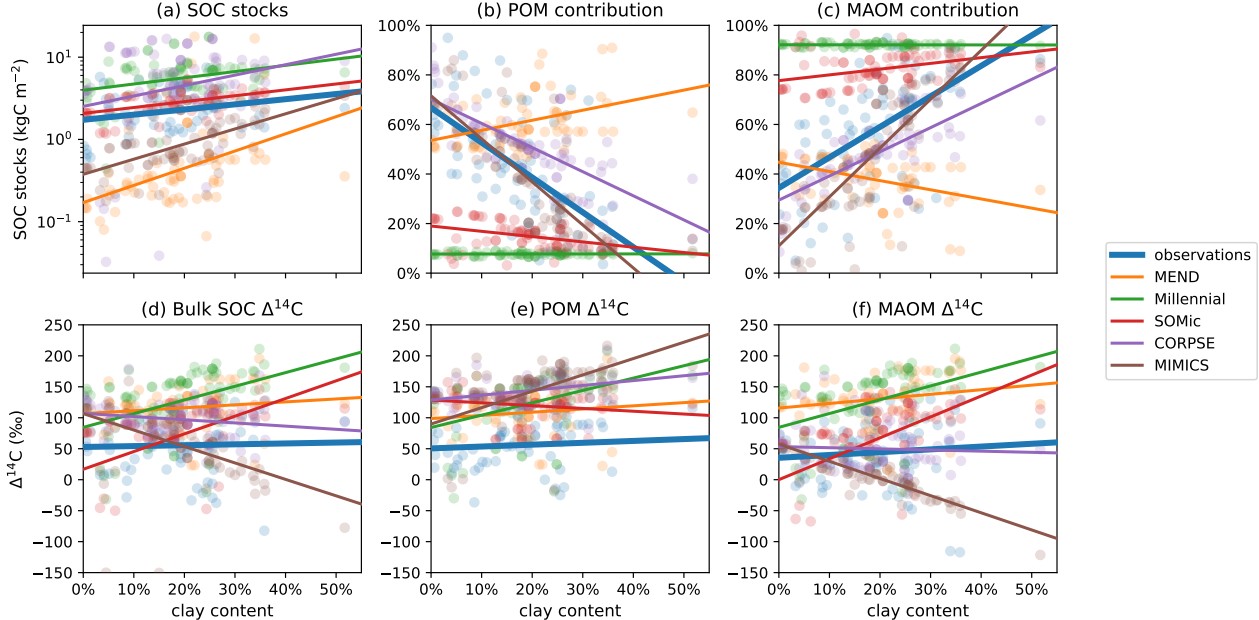

**Figure 8.** Relationships of observed and predicted carbon and $\Delta^{14}$C data with respect to clay content in the topsoil. Circles are datapoints, and lines are best linear fits through the points. CORPSE and MIMICS successfully reproduce the positive relationship between topsoil clay content and the observed MAOM contribution to the SOC stocks in subplot (c). However, in subplot (f), MIMICS has a strong negative correlation of MAOM $\Delta^{14}$C with clay content, unlike the observations, which do not show a correlation. The linear fit line of CORPSE in subplot (f) overlaps with that of the observations. Note that some extreme outliers are outside of the plotting range, and that we once again excluded the 1949 and 1978 samples for these plots. Separate plots for each individual model are provided in the Supplementary Material (Figures S.1.2.26–30).

## 4    Discussion

The comparison of topsoil $^{14}$C measurements with predictions by new-generation models reveals inaccuracies in the modeled time scales of carbon turnover and persistence in soils. Like the Earth System Models (ESMs) evaluated in He et al. (2016), most new-generation models do not correctly reproduce the $\Delta^{14}$C of bulk soil organic carbon (SOC) and they, too, may therefore be unsuitable for studying the effectiveness of soils as a net atmospheric $CO_2$ sink in the 21st century. The model biases in the partitioning of SOC between particulate organic matter (POM) and mineral-associated organic matter (MAOM) may also affect the accuracy of future projections. POM and MAOM have been shown to have different sensitivities to environmental variables such as temperature and are thus expected to react differently to a changing climate (Georgiou et al., 2024; Heckman et al., 2022; Hicks Pries et al., 2017; Kleber et al., 2011). Therefore, if models do not correctly partition SOC into POM and MAOM and misrepresent their $^{14}$C, they will probably produce inaccurate predictions of SOC dynamics under climate change.

We identify three likely reasons why the new-generation models generally underperform with $^{14}$C, and discuss how these problems could potentially be solved:

1. Insufficient datasets for the calibration of carbon turnover parameters,

2. Lack of a pool with very slow turnover to account for highly persistent SOC components,

3. Pools which do not capture the full range of SOC turnover rates.

The last point invites further research on the stability of the different constituents of SOC, and a discussion on the most effective way to partition SOC into pools which are more representative of the diversity of cycling rates and persistence of carbon in soils.

## 4.1 Insufficient calibration datasets

Our $^{14}$C results suggest that the new-generation models selected for this study overestimate some carbon turnover rates. The most extreme case is Millennial v2, which gives its micro-aggregate pool and mineral-adsorbed carbon pool turnover times of just a few months (see Appendix F). On the other hand, $^{14}$C-based studies find that the MAOM fraction, which includes micro-aggregates and mineral-adsorbed carbon, typically turns over on time scales of many decades or centuries (Gaudinski et al., 2000; Schrumpf and Kaiser, 2015; Van der Voort et al., 2017; Baisden et al., 2002). The overestimation of turnover rates may be due to inadequate or insufficient data for the calibration of the models' turnover parameters. Even though new-generation models can model measurable soil fractions such as POM and MAOM, they do not usually assimilate fraction-specific carbon and $^{14}$C data, probably because such data are currently very sparse. The only models in our evaluation to calibrate their parameters with fraction-specific carbon data are CORPSE (with data from only 2 soil profiles, according to Zhang et al., 2021, Table S1) and Millennial (as described in Abramoff et al., 2022), and none of them assimilated fraction-specific $^{14}$C data. Instead, new-generation models primarily rely on less informative bulk soil data, as well as some soil incubation results, for parameter optimization. However, as the dataset of fraction-specific carbon and $^{14}$C measurements is growing larger, new-generation models should start to take full advantage of the measurability of their pools and assimilate those highly informative data.

## 4.2 Lack of passive pool

Another explanation for the consistent overestimation of soil $\Delta^{14}$C by new-generation models is the inability of the models to account for the presence of highly persistent compounds in the soil, which negatively offset the bulk $\Delta^{14}$C. For example, some soils with a history of wildfires may contain a considerable fraction of pyrogenic carbon (Reisser et al., 2016; González-Domínguez et al., 2019), which is composed of highly durable aromatic compounds and can remain in soils over thousands of years (Eckmeier et al., 2009; Hajdas et al., 2007; Leifeld, 2008). Due to its longevity, pyrogenic carbon is depleted in $^{14}$C as a result of radioactive decay, bringing down the overall $\Delta^{14}$C of both POM (Van der Voort et al., 2017; Baisden et al., 2002) and MAOM (Soucémarianadin et al., 2019). In deeper soils, the $\Delta^{14}$C of SOC can be even further depleted due to a larger

proportion of petrogenic carbon, which is devoid of $^{14}$C (Grant et al., 2023; Van der Voort et al., 2019). Whereas the two major traditional SOC models explicitly account for such extremely old components with a "passive" pool (1000 year turnover time) in the Century model (Parton et al., 1987) and an "inert organic matter" pool (no turnover at all) in the RothC model (Coleman

and Jenkinson, 1996), the new-generation models effectively force virtually inert components to fit into their actively cycling carbon pools. By adding slow-turnover pools to account for highly persistent compounds such as pyrogenic carbon, the new-generation models would be able to lower the overall $\Delta^{14}$C of POM and MAOM, and more accurately reproduce the measured $^{14}$C data.

## 4.3 Search for more representative pools

Finally, the underperformance of the models with respect to $^{14}$C may also be due to a choice of pools which are not truly representative of the full spectrum of carbon turnover rates in soils. Whereas traditional models simply define the number and turnover rates of their SOC pools such that they can reproduce observed SOC dynamics while minimizing their degrees of freedom, new-generation models additionally need to make sure their pools are at once easily measurable and representative of the various time scales of SOC persistence. If a measurable fraction contains two or more components with very different

turnover rates, as is the case for the POM and MAOM fractions (von Lützow et al., 2007; Poeplau et al., 2018; Baisden et al., 2002), a model will not be able to correctly reproduce the fraction's $\Delta^{14}$C with one single carbon pool because it assumes a homogeneous turnover rate for the entire pool. Most new-generation models already address this problem by splitting the POM and MAOM fractions into multiple smaller subpools with contrasting turnover rates. For example, the SOMic model distinguishes between soluble and insoluble POM, and the MEND model between oxidizable and hydrolysable POM. Some

new-generation models subdivide the MAOM fraction into micro-aggregates and mineral-adsorbed carbon (e.g., Millennial model), or into an active layer of adsorbed DOC (dissolved organic carbon) and a more stable MAOM component (e.g., MEND model). However, these subpools might still not be homogeneous enough in their turnover rates for effective $^{14}$C simulations. Recent $^{14}$C studies determining the stability of MAOM under the action of peroxide oxidation show that it may be necessary to further split clay-sized MAOM into two measurable subpools which are decomposable or resistant to microbial

exo-enzymes (Schrumpf et al., 2021; Jagadamma et al., 2010; Poeplau et al., 2018). Within the POM fraction, the occluded light fraction could serve as an easily measurable proxy for the more persistent POM (Schrumpf et al., 2013; Wagai et al., 2009), and measurements of the pyrogenic carbon content (e.g., with hydrogen pyrolisis, as in González-Domínguez et al., 2019) could give clues on the size of the most persistent pool in the POM fraction. Finally, "continuous" SOC fractionation methods such as ramped pyrolysis oxidation (Stoner et al., 2023) could provide a much higher resolution of the SOC turnover

rate spectrum. However, the resulting measurable pools are more difficult to interpret in terms of their role in the soil carbon cycle, and their incorporation into mechanistic SOC models is therefore less straightforward. In order to correctly reproduce the time scales of SOC persistence and turnover, new-generation models may need a more detailed subdivision of the POM and MAOM fractions into more representative subpools, thus potentially increasing the number of simulated pools and degrees of freedom. However, as discussed in section 4.1, such an increase in model complexity must also be accompanied with an

expansion of the observational datasets, in particular fraction-specific isotopic measurements, for effective model calibration and validation.

## 4.4 Limitations of this study

The accuracy of our model evaluation is affected by several factors. Though we took care to accurately match the modeled pools to the measured fractions (see Appendix C), the correspondences are imperfect and further complicated by non-standardized
definitions and density cut-offs for the light and heavy fractions published on ISRaD. Nevertheless, this does not change the overall overestimation of soil $\Delta^{14}$C by most models. The use of forcing data from possibly inaccurate CESM2-LE and CCMI outputs with low spatial resolution may also affect the accuracy of our model evaluation. Furthermore, the $\Delta^{14}$C of the carbon inputs from the CESM2-LE product could be inaccurate, especially in soils with a thick organic layer, which pre-ages the carbon before it enters the mineral soil. However, the consistency and magnitude of the models' overestimation of the topsoil's
$\Delta^{14}$C with respect to observed data indicate that this overestimation is evidently a real pattern among the studied models. Finally, it is also important to note that our study only produces an incomplete picture of model performances on a global scale, since most of the measured datapoints represent North American and European forest ecosystems.

## 5 Summary

Despite their incorporation of the latest advances in soil sciences, new-generation soil organic carbon (SOC) models currently
face similar problems with predicting $^{14}$C as the traditional SOC models. The new-generation models' consistent overestimation of the $\Delta^{14}$C in both particulate organic matter (POM) and mineral-associated organic matter (MAOM) and their inaccurate partitioning of SOC between the POM and MAOM fractions suggest that these models underestimate the time scales of carbon storage in soils and might produce unreliable future predictions under climate change. To improve their predictions, new-generation models should take advantage of the measurability of their pools and calibrate their parameters with the rapidly
growing dataset of fraction-specific carbon and $^{14}$C measurements in addition to incubation and bulk soil data. They may also have to reconsider their model design and simulate carbon pools which better capture the full spectrum of carbon turnover rates present in the soils. In particular, the consideration of highly persistent SOC such as pyrogenic carbon could significantly improve $^{14}$C predictions. As more effective measurable pools are being discovered and the dataset of fraction-specific $^{14}$C data is expanding, new-generation soil models have the potential to eventually supersede the traditional SOC models employed by
ESMs if they take full advantage of the measurability of their pools and assimilate the available data.

*Code and data availability.* The source code to download the input data, run the models, and reproduce all the results presented in this manuscript and the supplementary material is available in our GitHub repository https://github.com/asb219/evaluate-SOC-models, published on Zenodo at https://zenodo.org/records/10575139 (Brunmayr, 2024).

## Appendix A: ISRaD data selection and processing

### A1 Derivation of LF data from fLF and oLF data

We calculate the $\Delta^{14}C$ and carbon contribution of the light fraction (LF) by combining the soil density fraction data of the free light fraction (fLF) and the occluded light fraction (oLF) from the International Soil Radiocarbon Database (ISRaD) (Lawrence et al., 2020). The fractional contribution of LF to the total soil organic carbon ($c^{LF}$) is calculated as the sum of the fLF and oLF contributions ($c^{fLF}$ and $c^{oLF}$, respectively), and the $\Delta^{14}C$ of LF is derived with a weighted average of the $\Delta^{14}C$ of fLF and oLF:

$$\Delta^{14}C^{LF} = \frac{c^{fLF} \cdot \Delta^{14}C^{fLF} + c^{oLF} \cdot \Delta^{14}C^{oLF}}{c^{LF}} , \tag{A1}$$

where $c^{LF} = c^{fLF} + c^{oLF}$.

### A2 Derivation of bulk data from LF and HF data

If the $\Delta^{14}C$ data for the bulk soil ($\Delta^{14}C^{bulk}$) are not available, we derive them with a weighted average of $\Delta^{14}C^{LF}$ and $\Delta^{14}C^{HF}$, the $\Delta^{14}C$ of the light fraction (LF) and heavy fraction (HF), respectively:

$$\Delta^{14}C^{bulk} = \frac{c^{LF} \cdot \Delta^{14}C^{LF} + c^{HF} \cdot \Delta^{14}C^{HF}}{c^{LF} + c^{HF}} , \tag{A2}$$

where $c^{LF}$ and $c^{HF}$ are the LF's and HF's relative contributions to the soil organic carbon stocks, respectively. Note that the sum $c^{LF} + c^{HF}$ is generally very close to 1, but not necessarily equal to 1, depending on the methods employed by the data producers.

### A3 Definition of topsoil and selection of profiles

We define the topsoil as at least the top $5\,cm$ and at most the top $10\,cm$ of the mineral soil, i.e., the interval from $0\,cm$ to $x\,cm$ depth such that $5 \leq x \leq 10$. All profiles in this study must have depth layers which fully span the topsoil without a gap. We only use layers whose top boundary is less than $5\,cm$ deep and whose bottom boundary is less than $10\,cm$ deep. For example, if a profile has layers 0-5 cm and 5-10 cm, we only use the 0-5 cm layer to represent the topsoil and discard the data from the 5-10 cm layer.

Examples of profiles we would choose for this study:

- Profile with layer 0-10 cm

- Profile with layers 0-3 cm and 3-8 cm

- Profile with layers 0-4 cm and 3-8 cm (overlapping is allowed)

Examples of profiles that we would have to reject:

- Profile with layer 0-15 cm (extends beyond 10 cm depth)

- Profile with topmost layer 1-8 cm (missing top 1 cm)

- Profile whose top two layers are 0-3 cm and 4-8 cm (gap between layers)

## A4 Derivation of topsoil data from layer data

The carbon and $^{14}$C data for the topsoil are derived by integrating over the layers comprising the topsoil. The total soil organic
carbon stocks in the topsoil (SOC) are found by summing the $SOC_\ell$ stocks in each layer $\ell$. If the $SOC_\ell$ data are not reported,
they are derived from the layer thickness $h_\ell$, soil bulk density $\rho_\ell$, and carbon concentration $C_\ell$ in each layer $\ell$:

$$SOC = \sum_\ell SOC_\ell = \sum_\ell h_\ell \rho_\ell C_\ell. \tag{A3}$$

In order to find the $\Delta^{14}C$ of bulk soil, light fraction (LF), and heavy fraction (HF) in the topsoil ($\Delta^{14}C^{\text{bulk}}$, $\Delta^{14}C^{\text{LF}}$, and
$\Delta^{14}C^{\text{HF}}$, respectively), as well as the LF and HF fractional contributions to the total carbon stocks in the topsoil ($c^{\text{LF}}$ and $c^{\text{HF}}$,
respectively), we take a weighted average over the layers $\ell$:

$$\Delta^{14}C^{\text{bulk}} = \sum_\ell SOC_\ell \cdot \Delta^{14}C_\ell^{\text{bulk}}/SOC \tag{A4}$$

$$\Delta^{14}C^{\text{LF}} = \sum_\ell SOC_\ell \cdot c_\ell^{\text{LF}} \cdot \Delta^{14}C_\ell^{\text{LF}}/(SOC \cdot c^{\text{LF}}) \tag{A5}$$

$$\Delta^{14}C^{\text{HF}} = \sum_\ell SOC_\ell \cdot c_\ell^{\text{HF}} \cdot \Delta^{14}C_\ell^{\text{HF}}/(SOC \cdot c^{\text{HF}}) \tag{A6}$$

$$c^{\text{LF}} = \sum_\ell SOC_\ell \cdot c_\ell^{\text{LF}}/SOC \tag{A7}$$

$$c^{\text{HF}} = \sum_\ell SOC_\ell \cdot c_\ell^{\text{HF}}/SOC \tag{A8}$$

If there are overlapping layers in the topsoil (e.g., a profile with layers 0-2 cm, 0-4 cm, and 3-10 cm), we integrate over depth
while averaging overlapping layers in the intervals where those layers overlap.

## Appendix B: Further information on model versions and implementations

The original source codes of all the selected model versions are openly available. By having direct access to the code with
which the model developers produced their results, we can be more confident that we test an implementation of the models as
intended by their respective authors.

Our final implementations of Millennial, CORPSE, MIMICS, and the $^{14}$C component of MEND are available as python
modules in our GitHub repository https://github.com/asb219/evaluate-SOC-models, published on Zenodo at https://zenodo.
org/records/10575139 (Brunmayr, 2024). For the carbon and nitrogen components of MEND, we compile the Fortran source
code from https://zenodo.org/records/11065513 (Wang and Brunmayr, 2024). Finally, we use the `install_url` function
of the `devtools` package in R (Wickham et al., 2022) to install SOMic as an R package directly from https://zenodo.org/
records/11068749 (Woolf and Brunmayr, 2023).

## B1  MEND

We use the MEND-new version of the MEND model as described in Wang et al. (2022). Our $^{14}$C re-implementation is based on the code from commit `92323c7` of the GitHub repository https://github.com/wanggangsheng/MEND (Zenodo publication: Wang, 2024). We forked the repository from that commit to https://github.com/asb219/MEND so that we could adapt the model input and output to our purposes. On our fork, the original version of MEND-new is released under tag name "MEND-new", and the version we used to produce our results is released under tag name "MEND-new-asb219" (Zenodo publication: Wang and Brunmayr, 2024). We use all the default model settings and the optimized parameter values provided in the Fortran namelist file `MEND_namelist.nml` in the repository. The pre-industrial soil carbon and nitrogen stocks are found by initializing the model with the default initial state from file `userio/inp/SOIL_ini.dat` and spinning up the non-isotopic carbon–nitrogen component of the model for 400 years with pre-industrial forcing data. The pre-industrial soil $^{14}$C levels are then found by running the $^{14}$C component of the model for another 1000 years, looping over the final year of the carbon–nitrogen spinup. The final states of the carbon–nitrogen and $^{14}$C spinups are then used for the initial condition of the final run of MEND over the 1850–2014 period. The model runs with hourly time steps and uses the forward Euler integration method.

## B2  Millennial

We use Millennial V2 with Michaelis-Menten kinetics as described in Abramoff et al. (2022). We re-implemented the model with $^{14}$C in Python based on the original R code in the https://github.com/rabramoff/Millennial repository released under the tag "v2", commit `e95bca9` (Zenodo publication: Abramoff and Xu, 2022). We used the model equations from file `R/models/derivs_V2_MM.R` in the repository and ran the model with the fitted parameter values from the file `Fortran/MillennialV2/simulationv2/soilpara_in_fit.txt` in the repository. The initial condition for both carbon and $^{14}$C stocks is found by first solving for a pre-industrial steady state (similarly to the model tutorial `R/simulation/model_tutorial.Rmd` in the repository), and then running the model from steady state for 200 years using time-varying pre-industrial forcing data featuring a seasonal cycle. The final state of that spinup is then used as the initial condition for the final run of the model over the 1850–2014 period. The model runs with daily time steps, and though the model tutorial uses the 4th order Runge-Kutta integration method, we integrate the equations simply with the forward Euler method, which is stable and precise enough with daily time steps.

## B3  SOMic

We use version 1.0 of the SOMic model as described in (Woolf and Lehmann, 2019). The original code is released under version "SOMic v 1.00" (commit `be34e56`) in the GitHub repository https://github.com/domwoolf/somic1 (Zenodo publication: Woolf, 2024). However, we had to fork the repository from commit `be34e56` to https://github.com/asb219/somic1 in order to fix a minor issue in the $^{14}$C implementation (see reason in Appendix E1), and to allow for distinct $^{14}$C values in the initial condition of each pool (previously, all pools were always initialized with the same $^{14}$C value). To produce our results, we used the version released under the tag "v1.1-asb219" in our fork (Zenodo publication: Woolf and Brunmayr, 2023). The model is

spun up for 50,000 years to get the initial carbon and $^{14}$C stocks. The model runs with monthly time steps and uses the forward Euler integration method.

## B4 CORPSE

The CORPSE model was originally described in Sulman et al. (2014). There are currently six publicly available versions of CORPSE owned by GitHub user https://github.com/bsulman. Since we are mostly interested in carbon dynamics, the lead developer Benjamin Sulman recommended we use the most up-to-date carbon-only implementation in https://github.com/bsulman/CORPSE-fire-response (commit `19ee2c7` released as version v1.0; Zenodo publication: Sulman, 2024a). We re-implemented CORPSE with $^{14}$C based on the equations in file `CORPSE_array.py` and using the parameter values from file `Whitman_sims.py` in that repository. However, the equation for the clay-related rate modifying factor is taken from file `code/CORPSE_integrate.py` in repository https://github.com/bsulman/CORPSE-N (commit `4a689ef` released as version v1.0; Zenodo publication: Sulman, 2024b), since the model seems to be working more reliably with that version of the equation. Like in Millennial, the initial condition is found by solving for a pre-industrial steady state and spinning up for 200 years from that steady state. If the solver is unable to find a steady state, the model is spun up for 10,000 years. The steady-state solution was found for all the profiles in this study. The model runs with daily time steps and uses the forward Euler integration method.

## B5 MIMICS

We use MIMICS-CN v1.0, as published in Kyker-Snowman et al. (2020), because the latest version of MIMICS (Wang et al., 2021; Wang, 2020) did not correctly implement $^{14}$C (see Appendix E2). The original R code of MIMICS-CN v1.0 is available on https://zenodo.org/records/3534562 (Kyker-Snowman, 2019). It already implements stable isotope tracers, but no radioactive isotopes such as $^{14}$C, so we re-implemented the model with $^{14}$C in python. Like for Millennial and CORPSE, we spin up for 200 years from the pre-industrial steady-state solution. If no steady state can be found, we spin up for 10,000 years. The steady-state solution was found for all the profiles in this study. The model runs with hourly time steps and uses the forward Euler integration method.

## Appendix C: Correspondences between pools and soil fractions

This section explains how we associate the simulated pools of each model with either the POM fraction ("particulate organic matter", corresponding to the "light fraction" resulting from density fractionation) or the MAOM fraction ("mineral-associated organic matter", corresponding to the "heavy fraction" resulting from density fractionation). We assume that the POM fraction is composed of fragmented and partially processed plant litter which is not stabilized in the soil matrix through mineral association. We assume that the MAOM fraction is composed of soil organic carbon which is enclosed in stable aggregates or strongly adsorbed to minerals. Since the live microbial biomass and dissolved organic carbon generally represent a small fraction of soil organic carbon, we can neglect them and assume they belong to neither POM nor MAOM.

See Table C1 for a summary of the correspondences between the modeled pools and the POM and MAOM fractions.

**Table C1.** Correspondences between simulated carbon pools and the POM fraction, MAOM fraction, or other carbon fractions. See Appendix sections C1–C5 for more information.

| Model | POM fraction | MAOM fraction | Other soil organic carbon pools | Litter pools |
|---|---|---|---|---|
| MEND | $POM_O$, $POM_H$ | MOM, QOM | DOM, $MB_A$, $MB_D$, $EP_O$, $EP_H$, EM, nosZ, norB, nirS & nirK, narG & napA, amoA & nxrA/B, nifH | |
| Millennial | POM | MAOM, Aggregate C | LMWC, Microbial Biomass | |
| SOMic | SPM, IPM | MAC | DOC, MB | |
| CORPSE | $SPC_u$, $CPC_u$ | $SPC_p$, $CPC_p$, $MN_p$ | $MN_u$, LMB | |
| MIMICS | $SOM_c$ | $SOM_p$ | $SOM_a$, $MIC_r$, $MIC_K$ | $LIT_m$, $LIT_s$ |

## C1 MEND

List of organic carbon pools in the MEND-new model by Wang et al. (2022) (model diagram in Figure C1):

– $POM_O$ and $POM_H$ (particulate organic matter decomposed by oxidative and hydrolytic enzymes, respectively).

– MOM (mineral-associated organic matter).

– QOM: "active layer of MOM" which can exchange carbon with DOM through adsorption and desorption (Wang et al., 2022).

– DOM (dissolved organic matter).

– $MB_A$ and $MB_D$ (active and dormant microbial biomass, respectively).

– $EP_O$, $EP_H$, EM, nosZ, norB, nirS, nirK, narG, napA, amoA, nxrA/B, nifH: various microbial exo-enzymes.

Note that the "Above-ground biomass", "Root biomass" and "Litter" boxes in the MEND model diagram in Figure C1 are not explicitly modeled as pools and therefore do not feature in the above list of organic carbon pools.

We assume that the POM fraction is composed of the $POM_O$ and $POM_H$ pools, and that the MAOM fraction is composed of the MOM and QOM pools. The DOM, $MB_A$, $MB_D$, and exo-enzyme pools belong to neither fraction.

## C2 Millennial

List of organic carbon pools in Millennial v2 by Abramoff et al. (2022) (model diagram in Figure C2):

– POM (particulate organic matter).

– Aggregate C: "stable microaggregates which remain after dispersion in the larger particle size fraction ($>$50–60 μm)" (Abramoff et al., 2022), so this corresponds to the coarse heavy fraction.

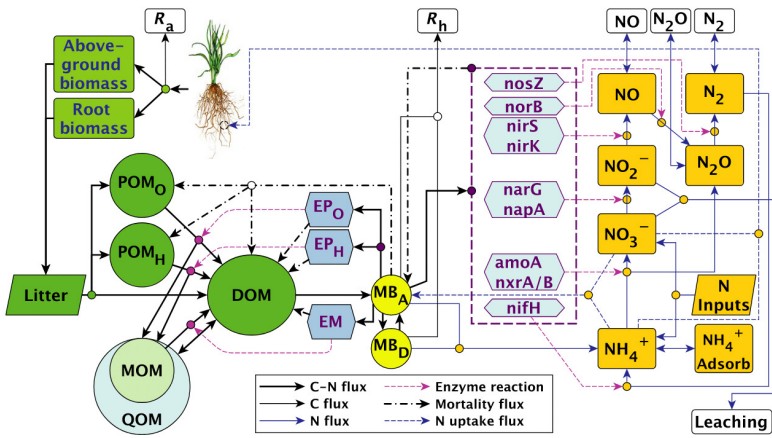

**Figure C1.** MEND-new model diagram. Source: Wang et al. (2022). Reuse permission received with Copyright Clearance Center license number 5691380194276.

– MAOM (mineral-associated organic matter): consists of organic matter associated to minerals through sorption (Abramoff et al., 2022).

  – Microbial Biomass: live microbial biomass.

  – LMWC (low molecular weight carbon): "LMWC could be analogous to dissolved organic C (DOC) if there is enough moisture in the soil matrix, and if we do not consider DOC molecules that are too large to be taken up by microbes"
(Abramoff et al., 2022).

We assume that the MAOM fraction is the sum of the Aggregate C and MAOM pools, and that the POM fraction is entirely composed of the POM pool. The Microbial Biomass and LMWC pools belong to neither fraction.

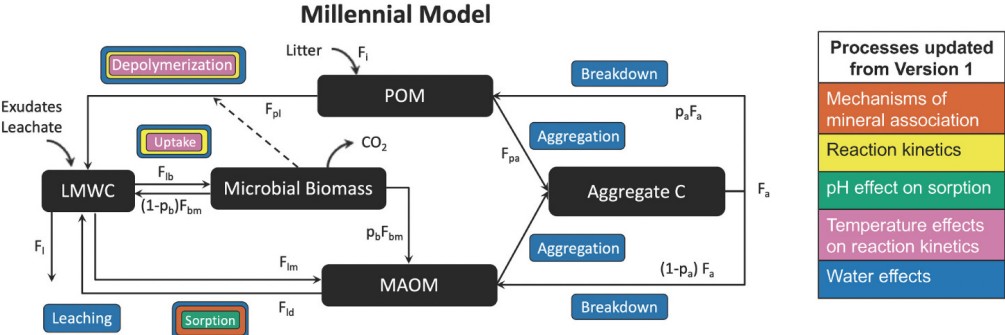

**Figure C2.** Millennial V2 diagram. Source: Abramoff et al. (2022). License: CC BY.

## C3   SOMic

List of organic carbon pools in SOMic 1.0 by Woolf and Lehmann (2019) (model diagram in Figure C3):

- SPM and IPM (soluble and insoluble plant matter, respectively).

- MAC (mineral-associated carbon): "mineral-sorbed or -occluded SOC" (Woolf and Lehmann, 2019).

- DOC (dissolved organic carbon).

- MB (microbial biomass).

We assume that the MAOM fraction is composed of the MAC pool, and the POM fraction is composed of the SPM and IPM
pools. The DOC and MB pools belong to neither fraction.

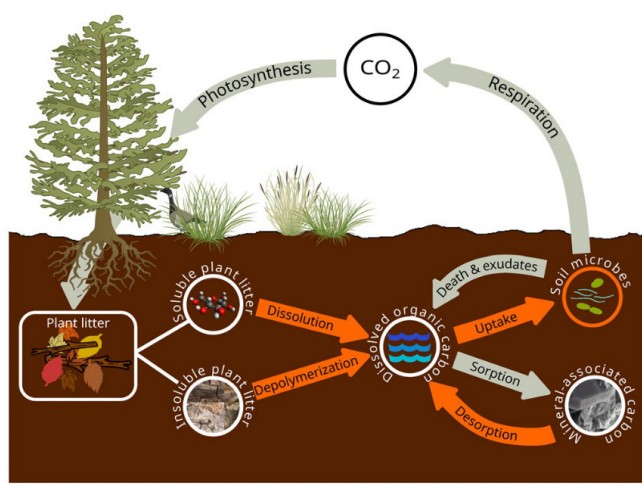

**Figure C3.** SOMic 1.0 diagram. Source: Woolf and Lehmann (2019). License: CC BY.

## C4   CORPSE

List of organic carbon pools in the CORPSE-fire-response version (Sulman, 2024a) of the CORPSE model, first published in
Sulman et al. (2014) and last updated in Moore et al. (2020) (model diagram in Figure C4):

- $SPC_u$, $CPC_u$, and $MN_u$ (Unprotected simple plant carbon, Unprotected complex plant carbon, and Unprotected microbe
necromass, respectively).

- $SPC_p$, $CPC_p$, and $MN_p$ (Protected simple plant carbon, Protected complex plant carbon, and Protected microbe necro-
mass): "protected organic matter is inaccessible to microbial decomposition through chemical sorption to mineral sur-
faces or occlusion within microaggregates" (Moore et al., 2020).

– LMB (live microbial biomass).

We associate the MAOM fraction with the $SPC_p$, $CPC_p$, and $MN_p$ pools, since they represent mineral-adsorbed and micro-aggregated carbon (Moore et al., 2020). We associate the POM fraction with the $SPC_u$ and $CPC_u$ pools, but not the microbial $MN_u$ pool, because POM is mostly composed of unprotected plant-derived carbon. The $MN_u$ and LMB pools belong to neither fraction.

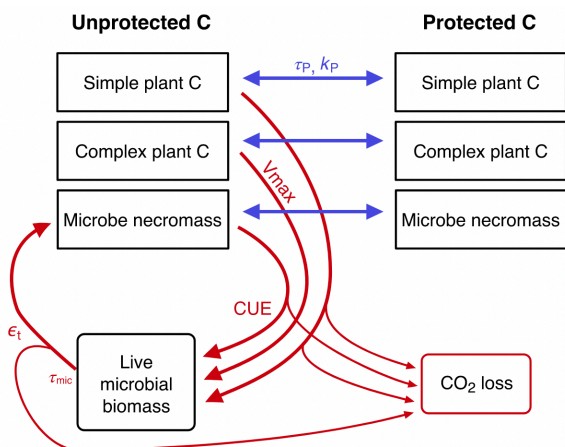

**Figure C4.** CORPSE diagram. Source: Moore et al. (2020). Reuse permission received with Copyright Clearance Center license number 5691370621010.

## C5 MIMICS

List of organic carbon pools in MIMICS-CN v1.0 by Kyker-Snowman et al. (2020) (model diagram in Figure C5):

– $LIT_m$ and $LIT_s$ (metabolic and structural litter, respectively): litter pools which are not considered part of soil organic matter.

– $SOM_p$ (physicochemically protected soil organic matter): "is primarily composed of microbial products that are adsorbed onto mineral surfaces" and is "analogous to heavy fraction or MAOM pools" (Kyker-Snowman et al., 2020).

– $SOM_c$ (chemically recalcitrant soil organic matter): "consists of decomposed or partially decomposed litter" and is "analogous to light fraction or POM pools" (Kyker-Snowman et al., 2020).

– $SOM_a$ (available soil organic matter): "the only SOM pool that is available for microbial decomposition; it contains a mixture of fresh microbial residues, products that are desorbed from the SOMp pool (e.g., Jilling et al., 2018), as well as depolymerized organic matter from the SOMc pool" (Kyker-Snowman et al., 2020). This pool is usually small and we associate it to neither POM nor MAOM.

– $MIC_r$ and $MIC_K$ ("low-efficiency, r strategist" microbes and "high-efficiency, K strategist" microbes, respectively): live microbial biomass pools.

According to Kyker-Snowman et al. (2020), the $SOM_c$ pool corresponds to the POM fraction, and the $SOM_p$ pool corresponds to the MAOM fraction. The $SOM_a$, $MIC_r$, and $MIC_K$ pools belong to neither fraction.

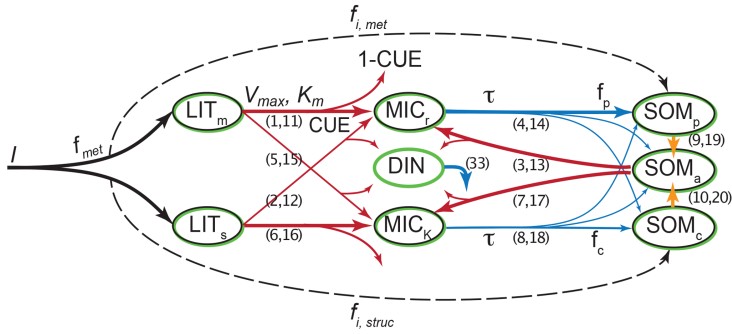

**Figure C5.** MIMICS-CN v1.0 diagram. Source: Kyker-Snowman et al. (2020). License: CC BY.

## Appendix D: Radiocarbon predictions with non-isotopic models

Among the new-generation models selected for this study, SOMic, MIMICS, and MEND have already implemented $^{14}$C. However, the most recent and only open-source version of MEND does not include $^{14}$C, and SOMic and MIMICS incorrectly implemented their $^{14}$C simulations (see Appendix E). Nevertheless, we can still produce $^{14}$C predictions with non-isotopic models by individually tracking the carbon fluxes at every time step and attaching a $^{14}$C signal to each flux. Since none of the models define an internal structure for their pools, we will assume by default that the pools are well-mixed, which means that the $\Delta^{14}$C of a pool's outflux is equal to the pool's $\Delta^{14}$C. This assumption is common practice for $^{14}$C modeling in soils (Sierra et al., 2017).

We run all of the selected models using the forward Euler method to advance from one time step to the next. The models either implicitly or explicitly produce the internal flux matrix $\Phi^i$ at each time step $i$, where $\Phi^i_{jk} \geq 0$ is the flux of carbon from pool $k$ into pool $j$ (with $j \neq k$), and $\Phi^i_{jj} \leq 0$ is the total outflux of carbon out of pool $j$ at time step $i$. They also define the external influx vector $I^i$ such that $I^i_j \geq 0$ is the influx of carbon entering the modeled system through pool $j$ at time step $i$. Matrix $\Phi$ contains all the fluxes between the pools and out of the system, and vector $I$ contains all the influxes of carbon from outside the system into the modeled pools. We can therefore find the carbon stocks $C^{i+1}_j$ of pool $j$ at time step $i+1$ based on the $\Phi^i$, $I^i$, and $C^i$ of the previous time step $i$:

$$C^{i+1}_j = C^i_j + I^i_j + \sum_k \Phi^i_{jk}, \tag{D1}$$

where the summation of internal fluxes $\Phi_{jk}^i$ is performed over all donor pools $k$ to get the total internal carbon flux into pool $j$ (when $k \neq j$), subtracted by the flux out of pool $j$ (when $k = j$).

Assuming the pools are well-mixed, we can now produce $^{14}$C predictions by tagging each flux $\Phi_{jk}$ with the $^{14}$C signal of pool $k$. We measure the $^{14}$C signal in terms of the unitless "absolute Fraction Modern" (FM$_{\mathrm{abs}}$) as defined in Trumbore et al. (2016), such that FM$_{\mathrm{abs}} = 1 + (\Delta^{14}\mathrm{C}/1000‰)$. The FM$_{\mathrm{abs}}$ is proportional to the $^{14}$C/$^{12}$C ratio normalized to a $\delta^{13}$C of $-25‰$ (Trumbore et al., 2016), and is thus proportional to the normalized ratio of $^{14}$C to total carbon ($^{14}$C/C), considering the negligible abundance of $^{14}$C compared to $^{12}$C and $^{13}$C. Therefore, if we know $F_j^i$, the FM$_{\mathrm{abs}}$ of pool $j$ at time step $i$, we can find $F_j^{i+1}$ at time step $i+1$ with the following equation (provided all the pools and the influx have comparable $\delta^{13}$C signatures):

$$F_j^{i+1} C_j^{i+1} = (1 - \lambda) F_j^i C_j^i + I_j^i F_{\mathrm{influx}}^i + \sum_k \Phi_{jk}^i F_k^i, \tag{D2}$$

where $C_j^{i+1}$ is given by equation (D1), $\lambda$ is the radioactive decay rate of $^{14}$C in units of inverse time step size, and $F_{\mathrm{influx}}^i$ is the FM$_{\mathrm{abs}}$ of the external carbon influx at time step $i$ given by the forcing data. We can then recover the $\Delta^{14}$C at each time step $i$ and for each pool $j$ with $(F_j^i - 1) \times 1000‰$.

## Appendix E: Incorrect or inaccurate $^{14}$C implementations

### E1 SOMic

The original implementation (available on Zenodo: Woolf, 2024) of the SOMic model (Woolf and Lehmann, 2019) does not produce accurate $^{14}$C predictions. Instead of working with the more typical $\Delta^{14}$C or absolute Fraction Modern (FM$_{\mathrm{abs}}$) units, this implementation tracks the $^{14}$C age, which we summarily define as $\mathrm{Age} = -\log(\mathrm{FM}_{\mathrm{abs}}) \lambda^{-1}$, where $\lambda$ is the radioactive decay rate of $^{14}$C. This causes complications when updating the $^{14}$C ages of the pools at each time step and when computing the total $^{14}$C age of the soil from the $^{14}$C ages of the individual pools. Indeed, to find the combined age $\mathrm{Age}_{\mathrm{A+B}}$ of pools A and B, the implementation of SOMic takes a weighted average over the ages, which is not entirely accurate:

$$\mathrm{Age}_{\mathrm{A+B}} = \frac{C_{\mathrm{A}} \mathrm{Age}_{\mathrm{A}} + C_{\mathrm{B}} \mathrm{Age}_{\mathrm{B}}}{C_{\mathrm{A}} + C_{\mathrm{B}}}, \tag{E1}$$

where $\mathrm{Age}_i$ and $C_i$ are the $^{14}$C age and the carbon stocks, respectively, of pool $i$. This weighted average formula is used to integrate the $^{14}$C ages of carbon fluxes into the pools at each time step on lines 154–160, and to compute the $^{14}$C age of the total soil on line 210 of file `src/SOMIC.cpp` (available on Zenodo: Woolf, 2024).

In order to prove that equation (E1) is inaccurate, let us derive how to correctly add the $^{14}$C ages of pools A and B. Let $^{14}C_i$ denote the $^{14}$C stocks and $C_i$ the total carbon stocks of pool $i$. Then, by conservation of mass, we have

$$^{14}C_{\mathrm{A+B}} = {}^{14}C_{\mathrm{A}} + {}^{14}C_{\mathrm{B}} \quad \text{and} \quad C_{\mathrm{A+B}} = C_{\mathrm{A}} + C_{\mathrm{B}} \quad \Rightarrow \quad \frac{^{14}C_{\mathrm{A+B}}}{C_{\mathrm{A+B}}} = \frac{^{14}C_{\mathrm{A}} + {}^{14}C_{\mathrm{B}}}{C_{\mathrm{A}} + C_{\mathrm{B}}}. \tag{E2}$$

Since the FM$_{\mathrm{abs}}$ is proportional to the $^{14}$C/C ratio (assuming pools A and B have similar $\delta^{13}$C signatures), the above is equivalent to

$$F_{\mathrm{A+B}} = \frac{C_{\mathrm{A}} F_{\mathrm{A}} + C_{\mathrm{B}} F_{\mathrm{B}}}{C_{\mathrm{A}} + C_{\mathrm{B}}}, \tag{E3}$$

where $F_i$ and $C_i$ are the $FM_{abs}$ and carbon stocks, respectively, of pool $i$. It follows that the combined $^{14}$C age of pools A and B is given by

$$\text{Age}_{A+B} = -\lambda^{-1} \cdot \log\left(\frac{C_A \exp\left(-\lambda \cdot \text{Age}_A\right) + C_B \exp\left(-\lambda \cdot \text{Age}_B\right)}{C_A + C_B}\right). \tag{E4}$$

Notice that equation (E1) is the first non-zero term of the above result's Taylor expansion around $\text{Age}_A = 0$, $\text{Age}_B = 0$. This means that equation (E1) works well for ages that are close to zero, i.e. when the $\Delta^{14}$C is close to zero. However, it fails to accurately predict the propagation of the bomb spike into the soil ecosystem in the latter half of the 20th century, as shown in Figure E1. While the error induced by the incorrect implementation exceeds 20‰ for the bulk soil $\Delta^{14}$C in the 1970s, the average error in the 2000s and 2010s is only around 10‰, which is relatively minor.

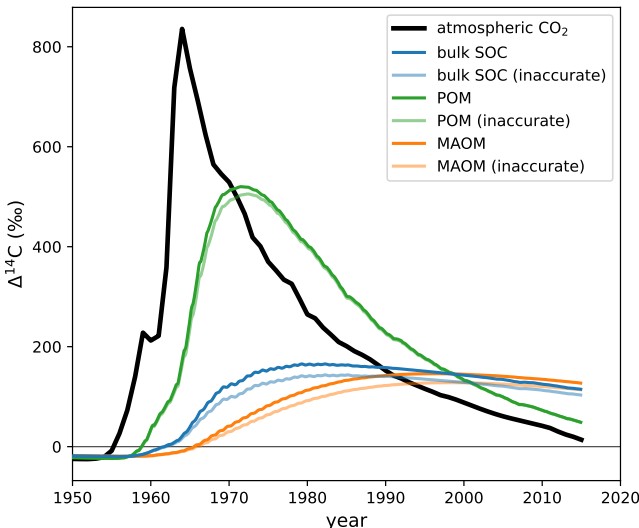

**Figure E1.** Comparison of $\Delta^{14}$C predicted by SOMic with the more and less accurate $^{14}$C implementations. For this example simulation, SOMic was run with forcing data corresponding to the top 5 cm of the mineral soil of the Bugac grassland site in Hungary, sampled in 2004 (Schrumpf et al., 2013). The atmospheric $\Delta^{14}$CO$_2$ of the Northern Hemisphere (Graven et al., 2017) is plotted for reference. The plotted model output data are available in the Supplementary Material (Table S.2).

## E2   MIMICS

The only MIMICS version already implemented with $^{14}$C is published in Wang et al. (2021), and the source code is available at https://data.csiro.au/collection/csiro:47942v1 (Wang, 2020). However, this $^{14}$C implementation is incorrect (see Figure E2).

The time evolution of the carbon stocks in MIMICS is given by function $f(C,t)$, which depends on the carbon stocks vector $C$ and time $t$. Function $f$ is implemented as subroutine `modelx` in the source file `vsoilmic05f_ms25.f90`. We can write function $f$ in terms of internal carbon transfer matrix $A$ and carbon influx vector $I$:

$$\mathrm{d}C/\mathrm{d}t = f(C,t) = A(C,t)C + I(t), \tag{E5}$$

where matrix $A(C,t)$ is a function of carbon stocks $C$ and time $t$, and vector $I(t)$ is time-dependent.

Then, following the same procedure which yielded equation (D2), we can derive the equation governing the evolution of the $^{14}$C stocks ($^{14}C$):

$$\mathrm{d}^{14}C/\mathrm{d}t = -\lambda^{14}C + A(C,t)^{14}C + {}^{14}I(t),\qquad\qquad\text{(E6)}$$

where $\lambda$ is the radioactive decay rate of $^{14}$C, and $^{14}I$ is the external influx of $^{14}$C.

However, in the $^{14}$C-implementation of MIMICS, the evolution of the $^{14}$C stocks is predicted with

$$\mathrm{d}^{14}C/\mathrm{d}t = -\lambda^{14}C + f({}^{14}C,t) = -\lambda^{14}C + A({}^{14}C,t)^{14}C + {}^{14}I(t).\qquad\qquad\text{(E7)}$$

The above equation is incorrect because, for this model, $A({}^{14}C,t) \neq A(C,t)$ when the pools have $\Delta^{14}C \neq 0‰$. This is especially problematic during the bomb-spike period, where $^{14}$C undergoes big changes while $C$ remains stable, causing $A({}^{14}C,t)$
to deviate significantly from $A(C,t)$. The incorrect implementation causes a strong attenuation of the $\Delta^{14}$C curves of the metabolic and structural litter pools (see Figure E2), which should more closely follow the atmospheric curve, considering the fast turnover rates of the litter pools. Another noticeable effect of the incorrect implementation, as seen in Figure E2, is that the $\text{SOM}_p$ pool (corresponding to the MAOM fraction) incorporated much more bomb-derived $^{14}$C than the $\text{SOM}_c$ pool (corresponding to the POM fraction) in the 1970s, which is highly improbable.

**Appendix F: Turnover times in the Millennial model**

In Millennial version 2 (Abramoff et al., 2022), the POM, MAOM, and Aggregate C pools exchange carbon with each other on the scale of a few months. The aggregate formation rate of the POM pool is between 0.012/day and 0.026/day ($k_{pa}$ in Table A1 of Abramoff et al., 2022), which translates to an average aggregation time of 1–3 months. Meanwhile, the optimized rate of aggregate formation for the MAOM pool is between 0.0038/day and 0.0052/day ($k_{ma}$ in Table A1 of Abramoff et al., 2022),
giving MAOM an average aggregation time of 6–8 months. The Aggregate C pool has a breakdown rate of around 0.02/day ($k_b$ in Table A1 of Abramoff et al., 2022), so aggregates have a turnover time of just 50 days. POM and MAOM exchange their carbon rapidly with the Aggregate C pool, which then redistributes the carbon back to the POM and MAOM pools in less than 2 months, on average. This means that, under the assumption of well mixed pools, the $^{14}$C signals of the POM, MAOM, and Aggregate C pools get homogenized within a couple years.

**Appendix G: Effect of sampling year on relationships between $^{14}$C and environmental parameters**

The results and analysis in section 3.3 on the dependency of observed and predicted $\Delta^{14}$C on environmental parameters could potentially be biased due to the different sampling years of soil profiles with different environmental parameters. While there is no strong relationship between soil temperature and the sampling year (Figure S.1.4), it turns out that most of the profiles with higher clay content (>20%) were sampled before 2005 and those with lower clay content (<20%) were sampled after

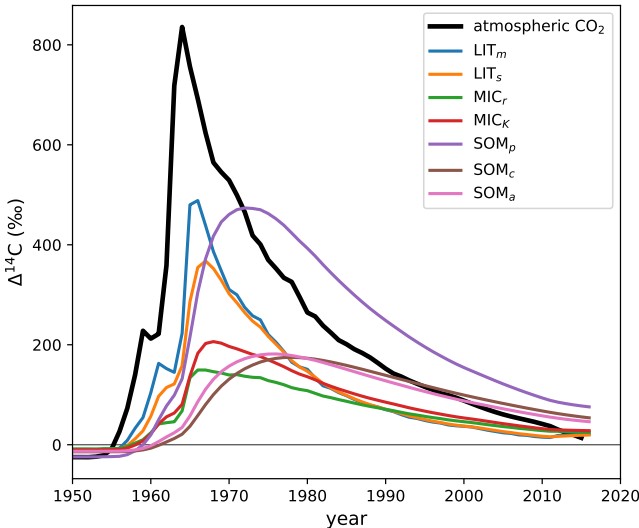

**Figure E2.** $\Delta^{14}$C output of MIMICS (Wang et al., 2021) with incorrect isotopic implementation. The model was run with the default parameters and forcing data published with the original source code (Wang, 2020). Our only modification to the source code was to output the pools' $^{14}$C and $^{12}$C stocks for each year. The atmospheric $\Delta^{14}$CO$_2$ of the Northern Hemisphere (Graven et al., 2017) is plotted for reference. MIMICS pool names: LIT$_m$, metabolic litter; LIT$_s$, structural litter; MIC$_r$, $r$-strategist microbes; MIC$_K$, $K$-strategist microbes; SOM$_p$, physically protected soil organic matter; SOM$_c$, chemically protected soil organic matter; SOM$_a$, active soil organic matter. The plotted model output data are available in the Supplementary Material (Table S.1).

2005 (Figure S.1.3). Even though the data shown in Figure 8 are only spanning a period of 18 years (1997–2015), the rapid changes in atmospheric $\Delta^{14}$CO$_2$ in the post-bomb period could mean that the regression lines of $\Delta^{14}$C against clay in subplots d–f are biased. We can attempt to remove this bias by "normalizing" the $\Delta^{14}$C data to the year 2000. The predicted $\Delta^{14}$C data are normalized simply by selecting the model output for 1 July 2000. The normalized $\Delta^{14}$C predictions for all models, profiles, and soil fractions are reported in Table S.5 (column names ending in "_14c_2000"). Normalizing the observed

$\Delta^{14}$C data, however, is highly problematic, especially in the context of this manuscript, because it requires the use of a simplistic soil carbon model. Following the normalization method used in Shi et al. (2020) and Heckman et al. (2022), we fit a steady-state linear one-pool model to the observed $\Delta^{14}$C data and then predict the $\Delta^{14}$C in the year 2000 with the fitted model. Table S.5 in the Supplementary Material lists the normalized $\Delta^{14}$C from the observed data (column names ending in "_14c_2000"), as well as the turnover rate of the one-pool model fitted with `scipy.optimize.minimize` in python

(column names ending with "_k", units of inverse years), and whether optimization terminated successfully (column names ending with "_success"), for each soil fraction and each profile. We then remade Figures 7 and 8 with all the $\Delta^{14}$C data normalized to the year 2000 (see Figures S.1.1.31 and S.1.1.25, respectively). Although normalization slightly shifted some of the $\Delta^{14}$C data, the slopes of the regression lines through the $\Delta^{14}$C data essentially remained the same. Therefore, our analysis and interpretation of the results presented in section 3.3 are likely not affected by the different sampling years.

*Author contributions.* Conceptualization: Brunmayr and Graven. Data curation, formal analysis, methodology, software, validation, visualization, and original draft preparation: Brunmayr. Investigation, and review and editing: Brunmayr, Graven, Hagedorn, Minich, Moreno Duborgel. Supervision: Graven. Funding acquisition: Graven and Hagedorn.

*Competing interests.* The authors declare no competing interests.

*Acknowledgements.* This study was supported by the Swiss National Science Foundation through the Sinergia scheme (grant no. 193770).

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
