# Peer review of "Radiocarbon analysis reveals underestimation of soil organic carbon persistence in new-generation soil models"

_Geoscientific Model Development, 2023_

## Author Response (AR2)

**Authors' Response**

2024-06-11

In addition to the changes outlined in our responses to the reviewers below (also posted as Author Comments in GMDD https://gmd.copernicus.org/preprints/gmd-2023-242/#discussion), we implemented the following changes in our revised submission:

1) After discussion with SOMic developer Dominic Woolf, we now use litterfall minus litter heterotrophic respiration instead of net primary production to estimate carbon inputs into the soil for the SOMic model. This significantly improved SOMic's predictions of SOC stocks in the topsoil, and it also improved its predictions of MAOM $^{14}$C.

2) We now perform a more accurate depth integration of the CESM2-LE output data by correctly weighting the data in the CESM2-LE depth layers which only partially overlap with the selected topsoil interval. The results did not change significantly.

3) We added labels to the tables and figures in the Supplementary Material for easier referencing in the manuscript, and we added "info" sheets with variable descriptions to the supplementary tables.

**Update 2024-06-17 (upload for typesetting and production)**
Contents did not change, only typesetting:
– added space before % and ‰ units
– replaced "Figure" with "Fig." and "Figures" with "Figs." when referencing figures
– slightly increased image resolution of SOMic model diagram (Fig. C3)
– removed hyphen in "$^{14}$C-implementation" (line 595)

**First Referee Comment**

RC1: 'Comment on gmd-2023-242', Jeffrey Beem Miller, 02 Feb 2024

New generation soil models include mechanistic pools, but lack validation. The authors of this work sought to address this research gap by evaluating 5 new generation soil C models using C and 14C data from SOM density fractions obtained from the International Soil Radiocarbon Database (ISRaD), focusing on topsoil. They found that the Δ14C of particulate organic matter (POM) pools were consistently overestimated (i.e., enriched) in the models relative to the observations, and the majority of the models overestimated Δ14C of mineral associated soil organic matter (MAOM) as well. The authors conclude that models could be improved through additional assimilation of measured 14C and C fraction data, which in turn may inform changes to model structures that would better fit available observations. As a specific recommendation, the authors point to the inclusion of an "inert" pool in the new generation models (a typical feature of 1st generation models such as CENTURY and RothC) to improve the fit to observed 14C data.

Overall, I found this manuscript to be a well-written and timely contribution to the field. However, there are a few technical issues with respect to the interpretation of the 14C data and what I feel to be some missed opportunities for comparing and contrasting the model results. I also have a concern about a fundamental premise of the work, i.e., that density fractions can be considered homogeneous pools. Perhaps this is more of a semantic issue, but I believe it warrants clarification.

I noted two key issues with the interpretation of the 14C data. First is the influence of the year of sampling, which is particularly relevant for assessing the relationship of Δ14C values to environmental variables with a linear regression approach. This source of potential bias can be addressed by normalizing Δ14C to a common year of sampling, as performed in Shi et al. (2020) and Heckman et al. (2022) (references listed below). The ISRaD R package provides a function for performing this normalization (ISRaD.extra.norm14c_year). More details on the specific concerns are outlined in the specific comment on lines 201-202. Second, the extremely strong dependence of 14C with depth in soils can lead to serious issues when drawing conclusions across studies with differing depth increments. Did you predict specific depths from the models? Did you spline the depth profiles from ISRaD (cf. Bishop et al., 1999; Malone et al., 2009; Sierra et al., 2024)? More information is needed here, and perhaps some adjustments to your approach.

Another area in which I think the study could be strengthened is further exploration of the nuances of the fraction data and the pool-specific model results. For example, I would be interested to know why the occluded particulate organic matter (oPOM) fraction was not explored further (which in some soils contains older C than the heavy fraction). Or, given the

current understanding that the heavy fraction is also likely a mixture of organic matter cycling across a range of time scales (and via distinct mechanisms), does the comparison of the observed data with more explicit pools of the models shed light on the environmental conditions under which MAOM is cycling relatively faster or slower (Heckman et al., 2018; Stoner, 2023)? There is some discussion of how the turn-over times assigned to the micro-aggregate and mineral adsorbed pools of the Millenial model are too fast (lines 235-236). Are there examples from the other models that can be compared and contrasted here? Can you draw some conclusions about which (putatively) measureable pools should be targeted to improve future models?

The authors identify the lack of a very slow pool in the models as a common issue across all of the models, which is a strong finding. However, the suggestion of simply adding an "inert" pool to compensate is not a novel concept, and in my view, does not help to move the field forward. Case in point, the slowest pool in the original RothC model (Jenkinson and Rayner, 1977) was "chemically stabilized soil organic matter" with a turnover of 1980 y. The "inert" pool was only introduced as a "deus ex machina" and less "elegant" solution (to quote the authors) for improving the fit of the model to observed 14C (Jenkinson, 1990). Furthermore, the suggestion that this inert material is equivalent to pyrogenic C is not convincing. Unfortunately, I do not think this is likely to be a useful solution either, given the known heterogeneity of pyrogenic C, which makes it ill-suited as a functional pool of SOM (unless further subdivided). The discussion (and analysis) could benefit from further discussion of how pyrogenic C may be incorporated into functional pools such as aggregates, or mineral-association, as well as the role of other potentially long-lived soil C pools.

Finally, the issue of whether empirical fractions can be considered modelable pools is at the heart of this work. Yet the terms "POM" and "MAOM" are used interchangeably for both empirical fractions and model pools throughout the paper, despite that fact that your results undermine this equivalence. I recognize that nomenclature is always a challenge, however, I think it needs to be made clearer that this is a hypothesis that is being tested, rather than an accepted fact. In light of your results, I would argue that a key addition to (or modification of) the list of likely reasons why the new-generation models underperform with respect to 14C (lines 225-229) is precisely the underlying heterogeneity of the measured fractions: i.e., the fact that these putative "measureable pools" are in fact a mixture of different pools of soil organic matter cycling at distinct rates, which in turn, are determined by different persistence mechanisms. Indeed, this paradigm is referenced briefly later in the discussion, especially with respect to the specific point about the role of pyrogenic C. Put another way, perhaps you could try to quantify what the benefits of density fractionation are for understanding soil organic matter dynamics, in light of the limitations that have been demonstrated here.

Is a key take-away from this work that we should continue performing density fraction, but simply take more care to separate charcoal from the light fractions? My understanding of the results is that this is not the case, and it seems like there is the potential here to provide more nuanced (and needed!) advice on how to move forward with modeling the measurable.

Line specific notes:

Ln 42-43: misleading, not just pedogenic oxides that form mineral-organic associations (Kleber et al., 2015). Clarify calcium bridging as a distinct mechanism in non-acidic soils (distinct from OM adsorption via surface charges of pedogenic oxides).

Ln 44: define residence time, cf. (Sierra et al., 2017)

Ln: 49: the Wagai reference here explicitly discusses the importance of mineral coating for the occluded light fraction. Most POM does not have mineral coatings.

Ln 51: can you elaborate on how this micro aggregate protection is related to occlusion or distinct from adsorption mechanisms?

Ln 52-54: This is a bit of an overstatement. Yes, there is the potential to measure the model pools, and it is true these models were designed, by and large, for the pools to be measureable. However, as this paper clearly identifies, much work remains to be done to make adequate empirical measurements of these new generation model pools.

Ln 72: Could also cite Metzler et al. (2020), who developed a mathematical approach for calculating $\Delta14C$ from any compartmental model, i.e., expressible as system of ODEs

Ln 97: I understand why you have made this nomenclature decision, but I am uncomfortable with the underlying assumption that these fractions are equal to the pools. A suggestion could be to add subscripts indicating measured vs modeled? Or please be clear that this is a hypothesis that is being tested here.

Ln 102: Please elaborate more on how you dealt with depth here (see notes above).

Ln 103: ISRaD contains substantially more density fraction 14C data than this. Can you clarify how you made your selections? Additionally, contrary to your statement in line 109, there are at least two studies with density fraction data from permafrost soils (O'Donnell et al., 2011; Gentsch et al., 2018).

Ln 158: Do these 14C data account for potential pre-aging in vegetation (Gaudinski et al., 2000; Joslin et al., 2006; Herrera-Ramírez et al., 2020)?

Ln 179: This is a bit confusing as written. He et al. (2016) found that ESMs consistently underestimated the age of soil C, which is also what you are suggesting here. But in terms of overestimating $\Delta14C$, that has a rather different interpretation when you are talking about pre-bomb $\Delta14C$ values below 0, as the results from He et al. show, versus modern era $\Delta14C$ values above 0, as you are showing. In the case of He et al., this indicated models were underestimating soil C ages by ~ hundreds of years. In your case it is much more ambiguous, and may only indicate differences of tens of years. Perhaps you could reword this to clarify what you mean.

Ln 192: Why was this site selected? Is it representative?

Ln 195: perhaps refer to the appendix?

Ln 201-202: This explanation needs some more interpretation (and is probably more suited the discussion). Given the large amount of change observed in atmospheric $\Delta14C$ over the period 1997-2015 (103 to 14 per mil) (Graven et al., 2017), these trends are ambiguous. For example, it could be just as likely that POM $\Delta14C$ would decrease with increasing temperature (as it does in some of the models) as the pool is cycling faster and is therefore closer to the declining atmospheric signal. The predicted trend should be a function of both the partitioning of C into the pool (its size) and its turnover rate.

Ln 207-208: Clarify observed vs. modeled?

Ln 217: Again, careful with the comparison to He et al. 2016. Overestimation of $\Delta14C$ means something very different in the context of that study versus this one (see note at line 179)

Ln 219: Perhaps "partitioning" instead of "repartition"?

Ln 230: Please elaborate more on the effectiveness of this distinction.

Ln 240: Perhaps "putatively" measureable pools?

Ln 249: "practically inert" is perhaps misleading here. The intent is clear, but many studies have confirmed that there is no "inert" soil C, only soil C that is inaccessible or either stoichiometrically or energetically unfavorable for microbes to consume.

Ln 252: "extreme longevity" this is not technically correct, as pyrogenic C is known to exhibit a wide range of 14C values with inferred mean residence times from decadal to centennial timescales (cf. Fig. 1, Schmidt et al., 2011)

Ln 254: consider citing (Grant et al., 2023)

Ln 255: And indeed, this was due to these models being informed by soil 14C measurements, e.g., Jenkinson and Rayner, 1977.

Ln 258-260: Reintroducing an "inert" pool seems to be contrary to the goals of these new models of having measurable pools in the models, as outlined by Lehmann and Kleber (2015) and also creates instability issues when searching for analytical model solutions (Sierra and Mueller, 2015).

Ln 266-268: Yes, this is a key point!

Ln 270: I would be cautious in assuming that pyrogenic C itself is a functional pool. The bigger issue here is that it violates the premise that density fractionation yields homogenous pools.

Ln 285-287: As mentioned previously, pre-aging of C in vegetation is not only an issue with thick O horizons.

Ln 293: Please reword.

Ln 371-372: This contradicts the definition of POM as supplied in the introduction.

**Response to First Referee Comment**

Thank you very much for your thorough and in-depth review!

You rightly point out that the terminology can cause some confusion and misunderstandings. To avoid the implication of a perfect equivalence between the density fractions and the modeled POM and MAOM, we will rename the "measured POM fraction" to "light fraction" and the "measured MAOM fraction" to "heavy fraction". This will also remove any possible confusion that one of the premises of our work is "that density fractions can be considered homogeneous pools". The density fractions are not homogeneous, and the modeled POM and MAOM are only homogeneous when they are each modeled as a single pool. In many models, however, POM and MAOM are each composed of two or more distinct pools, so POM and MAOM are not homogeneous in those models. We recognize that both the homogeneous carbon pools and the POM and MAOM fractions being called "pools" in our manuscript presents one more source of confusion. We will therefore replace "(modeled) POM pool" with "(modeled) POM fraction", and "(modeled) MAOM pool" with "(modeled) MAOM fraction" throughout the manuscript. The individual homogeneous pools of the models will still be called "pools", as before.

**Updated terminology:**
– light fraction (LF) and heavy fraction (HF): soil density fractions which approximate the POM and MAOM fractions, respectively.
– POM and MAOM fractions: conceptual soil fractions which can be modeled by new-generation models, where each fraction is represented by one or more homogeneous pools.
– Pools: homogeneous carbon pools in models.

We agree that the differences in the sampling years introduce potential biases in the relationships of 14C vs. temperature and 14C vs. clay content in section 3.3. Your suggestion is to remove this bias by normalizing the 14C values to a common year as in Shi et al. (2020) and Heckman et al. (2022). This normalization method works by fitting a steady-state linear 1-pool model to the measured 14C datapoint in order to then predict what the 14C would have been if the sample had been taken in a different year. We implemented the normalization to the year 2000 and found that the 14C data gets shifted but that the slopes of the regression lines do not change, and that the sampling year bias therefore does not affect our analysis of the results. We added a new appendix section about the sampling year bias and the normalization method, and we added figures and tables with the normalized results in the Supplementary Material. However, we decided to keep all the results in the main text of the manuscript un-normalized. This is because it would be very difficult for us to justify the use of a simplistic 1-pool model for normalization, considering that our study focuses on complex non-linear, non-steady-state, multi-pool models, and especially in the context of our statements that it takes more than 1 pool to correctly predict 14C. Furthermore, for many of the 14C data points selected for our study, the normalization method can produce 2 possible values for the 14C in the common year (similar to Figure 15a in the supplement of Shi et al., 2020), and it would again be difficult to justify which of the 2 values we choose.

We run the models only for the topsoil, which we treat as one single homogeneous layer, so we do not have model predictions for specific depths. Depending on the profile, the topsoil data is for a layer with depth bounds varying between 0-5 cm and 0-10 cm. As you say, due to "the extremely strong dependence of 14C with depth", it is not entirely appropriate to compare ∆14C values within a dataset containing a mixture of 0-5 cm layers and 0-10 cm layers. But one could argue that using arbitrary depth intervals instead of horizon boundaries, or comparing data from different soil types would introduce an even bigger bias than that introduced by inconsistent depths across profiles. We did not spline the depth profiles from ISRaD because we decided that we wanted to compare the model outputs directly to the measurements as reported in ISRaD, without interpolating, extrapolating, or otherwise significantly manipulating the measured data. As a result, we had to reject the soil profiles whose layers only partially overlap with the topsoil (which we define as min. 0-5 cm and max. 0-10 cm). For example, we would not use a profile if its top layer was 0-25 cm or 1-7 cm, or if its two top layers were 0-3 cm and 3-15 cm. For most of our selected profiles, the topsoil is spanned entirely by one single layer (e.g., 0-7 cm). In cases where the topsoil is composed of multiple layers (e.g. 0-4 cm and 4-10 cm), we calculate the topsoil 14C, LF% and HF% with a weighted average. We explained this some more in the main text and also in a new Appendix section, which also includes the formulas for the weighted averages. See also our response to your comment on Line 103.

In section 4.3, we discuss different measurable subfractions of POM and MAOM which could be incorporated in new-generation models (e.g., add a MAOM subfraction that is more resistant to microbes, and is "measurable" as the residual fine HF after application of peroxide, line 275). But we agree that there are definitely more new fractionation methods worth mentioning. As you suggested, we will now additionally talk about the possibility of using oPOM as an easily measurable proxy for a more persistent component of POM. We will make sure to emphasize that new-generation models are already aware that a simple distinction between POM and MAOM pools is not sufficient to cover the wide range of carbon cycling rates in soils, so they often further split them into faster and slower subpools. However, their choice of subpools (e.g. MEND's oxidizable and hydrolysable POM pools) may still not be good enough (for 14C predictions), so they need to consider different subpools or split their subpools even further. For example, Millennial can divide its "MAOM" pool (which is associated with the measured fraction "fine HF")  into more or less resistant components and associate the resistant component with the "residual fine HF" (mentioned line 275). This of course would increase model complexity, which would again require more data to constrain the model parameters (as discussed in section 4.1).

**Line-specific comments**

*Ln 42-43: misleading, not just pedogenic oxides that form mineral-organic associations (Kleber et al., 2015). Clarify calcium bridging as a distinct mechanism in non-acidic soils (distinct from OM adsorption via surface charges of pedogenic oxides).*

Now reads as "stabilized by interactions with reactive soil mineral surfaces of pedogenic oxides or phyllosilicates through cation bridging, electrostatic interactions, or the formation of inner- and outer-sphere complexes". I cited Kleber et al. (2015) and Rowley et al.'s "Calcium-mediated stabilisation of soil organic carbon" paper (2018).

*Ln 44: define residence time, cf. (Sierra et al., 2017)*
Replaced with "persistence". Everywhere else (lines 202 and 213), we replaced "residence time" with "turnover time", which is preferred by Sierra et al. (2017).

*Ln: 49: the Wagai reference here explicitly discusses the importance of mineral coating for the occluded light fraction. Most POM does not have mineral coatings.*
Wagai et al. (2009) show that the free light fraction has mineral coatings too, perhaps to a lesser degree than the occluded light fraction, but still clearly visible (and explicitly labeled) in their Figure 2d. We replaced "which are usually covered" with "which may be covered".

*Ln 51: can you elaborate on how this micro aggregate protection is related to occlusion or distinct from adsorption mechanisms?*
We acknowledge that adsorption occurs inside micro-aggregates as well. We rephrased this sentence to: "On the other hand, the MAOM fraction contains organic matter chemically adsorbed onto reactive mineral surfaces, or stabilized by occlusion or adsorption inside micro-aggregates formed around sand, silt, or clay particles".

*Ln 52-54: This is a bit of an overstatement. Yes, there is the potential to measure the model pools, and it is true these models were designed, by and large, for the pools to be measureable. However, as this paper clearly identifies, much work remains to be done to make adequate empirical measurements of these new generation model pools.*
Replaced "the POM and MAOM pools of the new-generation models can be operationally defined…" with "the POM and MAOM fractions simulated by new-generation models are designed to be "measurable": they can be operationally defined…".

*Ln 72: Could also cite Metzler et al. (2020), who developed a mathematical approach for calculating ∆14C from any compartmental model, i.e., expressible as system of ODEs*
Line 72 states that even when new-generation soil models implement 14C, they rarely calibrate their parameters with 14C observations. We do not see the direct relevance of the Metzler et al. (2020) study here.

*Ln 97: I understand why you have made this nomenclature decision, but I am uncomfortable with the underlying assumption that these fractions are equal to the pools. A suggestion could be to add subscripts indicating measured vs modeled? Or please be clear that this is a hypothesis that is being tested here.*
We decided to change the nomenclature.

*Ln 102: Please elaborate more on how you dealt with depth here (see notes above).*
See our response to line 103.

*Ln 103: ISRaD contains substantially more density fraction 14C data than this. Can you clarify how you made your selections? Additionally, contrary to your statement in line 109, there are at least two studies with density fraction data from permafrost soils (O'Donnell et al., 2011; Gentsch et al., 2018).*

Yes, there are a lot more density fraction 14C data in ISRaD (including for permafrost), but they are not in what we defined to be "topsoil". We selected all the profiles in ISRaD which have density fraction data and whose depth layers can be combined to completely cover but not exceed the 0-x cm interval, where x is at least 5 cm and at most 10 cm. Many profiles were rejected because their topsoil layer exceeded the maximum limit of 10 cm, e.g. their top layer was 0-15 cm or 0-25 cm. Some profiles were rejected because they missed a centimeter between their layers, e.g. they had 0-4 cm and 5-10 cm but missed 4-5 cm. The reason why we chose to not go deeper than 10 cm is because this allows us to neglect the effects of vertical mixing and instead focus on the pools and their turnovers. We now explain this more clearly in the manuscript.

*Ln 158: Do these 14C data account for potential pre-aging in vegetation (Gaudinski et al., 2000; Joslin et al., 2006; Herrera-Ramírez et al., 2020)?*

Yes, the carbon (incl. 14C) in the litter pools of CESM2 first passes through the vegetation pools. We added this sentence to clarify: "These litter $\Delta$14C data account for the pre-aging of carbon in vegetation (Herrera-Ramírez et al., 2020; Solly et al., 2018) because the litter carbon first passes through the vegetation pools in the land module of CESM2 (CLM5, Lawrence et al., 2019)."

*Ln 179: This is a bit confusing as written. He et al. (2016) found that ESMs consistently underestimated the age of soil C, which is also what you are suggesting here. But in terms of overestimating $\Delta$14C, that has a rather different interpretation when you are talking about pre-bomb $\Delta$14C values below 0, as the results from He et al. show, versus modern era $\Delta$14C values above 0, as you are showing. In the case of He et al., this indicated models were underestimating soil C ages by ~ hundreds of years. In your case it is much more ambiguous, and may only indicate differences of tens of years. Perhaps you could reword this to clarify what you mean.*

We agree that our results are not directly comparable to those of He et al. (2016), especially because He et al. look at the top 1 meter and we only look at the top 5 or 10 cm. We rephrased this sentence to: "This is reminiscent of the ESMs' 14C predictions from He et al. (2016), which also overestimate the $\Delta$14C of SOC and underestimate its variability, though to a different extent and over a larger depth interval (top 1 m instead of the top 5 or 10 cm of the mineral soil)." In the following sentence, we use cautious instead of forceful/assertive wording: "our results could suggest" and "may be facing similar issues".

Side note: When running the new-generation models for deeper depth intervals, we actually do see underestimations of soil C ages by ~100s of years, just like for the ESMs. However, we decided against showing results for deeper soils because the discrepancies between predictions and observations could then be simply blamed on the treatment of vertical mixing of soil carbon (especially when most models are not vertically resolved). That would take away

from the discussion of the model design (i.e., the choice of modeled pools and fluxes) of new-generation models, which was our main focus, so we decided to only look at the topsoil, where vertical mixing plays a relatively minor role in the 14C dynamics.

*Ln 192: Why was this site selected? Is it representative?*
Yes, we edited the sentence to specify "a representative example of the model biases". This site was selected because it's representative (with reference to the boxplots of Figure 5): MEND, Millennial and SOMic overestimate the Δ14C of both POM and MAOM, meanwhile CORPSE gets the Δ14C of MAOM right but overestimates the Δ14C of POM and bulk soil, and MIMICS underestimates the Δ14C for MAOM and overestimates it for POM but correctly predicts the bulk Δ14C.

*Ln 195: perhaps refer to the appendix?*
Yes.

*Ln 201-202: This explanation needs some more interpretation (and is probably more suited the discussion). Given the large amount of change observed in atmospheric Δ14C over the period 1997-2015 (103 to 14 per mil) (Graven et al., 2017), these trends are ambiguous. For example, it could be just as likely that POM Δ14C would decrease with increasing temperature (as it does in some of the models) as the pool is cycling faster and is therefore closer to the declining atmospheric signal. The predicted trend should be a function of both the partitioning of C into the pool (its size) and its turnover rate.*
We removed this part of the sentence: "possibly due to shorter carbon turnover times in warmer soils".

*Ln 207-208: Clarify observed vs. modeled?*
Yes.

*Ln 217: Again, careful with the comparison to He et al. 2016. Overestimation of Δ14C means something very different in the context of that study versus this one (see note at line 179)*
Rephrased: "Like ESMs, most new-generation models do not correctly reproduce the Δ14C of bulk soil organic carbon (SOC) and they, too, may therefore be unsuitable for studying the effectiveness of soils as a net atmospheric CO2 sink in the 21st century (He et al., 2016)."

*Ln 219: Perhaps "partitioning" instead of "repartition"?*
Yes.

*Ln 230: Please elaborate more on the effectiveness of this distinction.*
Replaced [["The last point raises questions on the effectiveness of the new-generation models and the POM–MAOM distinction as a whole. This invites further research on the stability of the different constituents of SOC and a discussion on the most effective way to partition SOC into representative measurable pools."]] with [["The last point invites further research on the stability of the different constituents of SOC, and a discussion on the most effective way to partition SOC

into pools which are more representative of the diversity in the cycling rates and persistence of carbon in soils."]]
This is further discussed in section 4.3 of the manuscript.

*Ln 240: Perhaps "putatively" measureable pools?*
We were not sure that the meaning of this word would be clear to all readers so we have not added it. We replaced "Even though new-generation models have measurable pools" with "Even though new-generation models can model measurable soil fractions such as POM and MAOM".

*Ln 249: "practically inert" is perhaps misleading here. The intent is clear, but many studies have confirmed that there is no "inert" soil C, only soil C that is inaccessible or either stoichiometrically or energetically unfavorable for microbes to consume.*
Replaced with "highly persistent".

*Ln 252: "extreme longevity" this is not technically correct, as pyrogenic C is known to exhibit a wide range of 14C values with inferred mean residence times from decadal to centennial timescales (cf. Fig. 1, Schmidt et al., 2011)*
Removed the word "extreme".

*Ln 254: consider citing (Grant et al., 2023)*
Yes.

*Ln 255: And indeed, this was due to these models being informed by soil 14C measurements, e.g., Jenkinson and Rayner, 1977.*
Agreed.

*Ln 258-260: Reintroducing an "inert" pool seems to be contrary to the goals of these new models of having measurable pools in the models, as outlined by Lehmann and Kleber (2015) and also creates instability issues when searching for analytical model solutions (Sierra and Mueller, 2015).*
Replaced "By creating a passive pool to account for inert compounds" with "By adding slow-turnover pools to account for highly persistent compounds".

*Ln 266-268: Yes, this is a key point!*
Agreed.

*Ln 270: I would be cautious in assuming that pyrogenic C itself is a functional pool. The bigger issue here is that it violates the premise that density fractionation yields homogenous pools.*
Replace "they miss the most recalcitrant POM pool of pyrogenic carbon" with "they do not account for the most recalcitrant POM subpools, including pyrogenic carbon".

*Ln 285-287: As mentioned previously, pre-aging of C in vegetation is not only an issue with thick O horizons.*

CESM2 does pre-age the C in vegetation (which we now explain in the Methods section, see our response for line 158), but does not consider potentially thick O horizons (as in some forest soils).

*Ln 293: Please reword.*
Replaced "new-generation soil organic carbon (SOC) models currently show similar discrepancies with 14C data as the traditional SOC models" with "new-generation soil organic carbon (SOC) models currently face similar problems with predicting 14C as the traditional SOC models".

*Ln 371-372: This contradicts the definition of POM as supplied in the introduction.*
We believe that this is a misunderstanding that arose from our mentioning a "thin mineral coating" on some of the POM in the Introduction (see our response to line 49). "Mineral association" in the definition of Appendix B refers to strong bonds to mineral surfaces and occlusion in stable aggregates which would protect the C from decomposition. That's not the case for the thin mineral coating mentioned in the Introduction.

**Second Referee Comment**

RC2: 'Comment on gmd-2023-242', Anonymous Referee #2, 18 Feb 2024

The authors compile a suite of complex "new generation" soil carbon models that have either implemented radiocarbon into their model structure or have the ability to do so. They fix errors in previous 14C implementation in 2 models and run all models with similar inputs at a set of sites where soil D14C of soil fraction data has been collected and stored in the ISRAD Database. They find that there is generally an overestimation of soil D14C in these models, indicating that they are underestimating the degree of persistence of soil carbon. Overall, this is a good contribution to the soil carbon model community, fixing inaccuracies in 2 models and highlighting that the problems in carbon turnover are not solved by just adding more complexity.

For the measured data from ISRaD, the authors should consider adding sites that have just bulk carbon density and D14C data, even if they do not have the size fraction D14C data, to test if the consideration of more sites (spatially and temporally) changes the result.

While the figures are generally well-designed and easy to understand, I think it would be interesting to see the spatial patterns of how well these models are doing. I think one more figure of the spatial pattern of bulk soil carbon and bulk D14C RMSEs for specific sites may be helpful in this. Another metric that should be considered is the model RMSE of the difference between POM and MAOM D14C.

Line Comments:

87: "used" or "use" instead of "will use"

137-138: Details of the spinup of these models should be added to this section instead of the Appendix because the spinup time can significantly affect the D14C values. Also concerning the spinup, details in the Appendix are needed on the percentage of sites in each model that reached equilibrium "solver" values vs. the base 4000-5000 year periods.

192: Figure 6 makes me wonder if plotting the difference between the D14C values of MOAM and POM for all sites might be beneficial in model comparison. This figure is making it seem like the MIMICS model might be getting the bulk D14C right for the wrong reasons, but it is only one site. You could add this metric to Figure 5 and add more discussion of this difference.

505: Details are needed on how much this difference in the original MIMICS D14C implementation affects bulk, POM and MAOM D14C values in the new implementation. Something similar to line 491 for the SOMic model should be fine.

Citation: https://doi.org/10.5194/gmd-2023-242-RC2

**Response to Second Referee Comment**

Thank you very much for your review!

*For the measured data from ISRaD, the authors should consider adding sites that have just bulk carbon density and D14C data, even if they do not have the size fraction D14C data, to test if the consideration of more sites (spatially and temporally) changes the result.*
We did consider also running the models on all the soil profiles in ISRaD with bulk 14C data even when they don't have density fraction data. However, some of the models use up significant resources to run (the MEND model, for example, writes 5GB of data to disk for each model run), making this endeavor not as feasible, especially when trying to ensure the reproducibility of our results. Since the main focus of our study is about making use of isotopic data from measurable soil fractions for model validation, we prefer to keep the results focussed on the profiles that do have density fraction data.

*While the figures are generally well-designed and easy to understand, I think it would be interesting to see the spatial patterns of how well these models are doing. I think one more figure of the spatial pattern of bulk soil carbon and bulk D14C RMSEs for specific sites may be helpful in this. Another metric that should be considered is the model RMSE of the difference between POM and MAOM D14C.*
We agree, it would be very interesting to also see how the model errors depend on the environmental parameters to see in what types of environment models perform better (or worse). We produced plots of the model bias and absolute error versus temperature and clay content. These will be added to the Supplementary Material. Please see our response to line comment 192 about having the difference between POM and MAOM D14C as a model performance metric.

**Line comments**

*87: "used" or "use" instead of "will use"*
We replaced "will use" with "use".

*137-138: Details of the spinup of these models should be added to this section instead of the Appendix because the spinup time can significantly affect the D14C values. Also concerning the spinup, details in the Appendix are needed on the percentage of sites in each model that reached equilibrium "solver" values vs. the base 4000-5000 year periods.*
We add this sentence in the main text before pointing to Appendix A for more details: "In practice, the carbon-nitrogen component of the MEND model is spun up for 400 years and its 14C component for 1000 years, the SOMic model is spun up for 5000 years, and the remaining three models are spun up for 200 years from their pre-industrial steady-state solution." In Appendix A, we added short statements that the steady-state solution was always found for the profiles selected for this study.

*192: Figure 6 makes me wonder if plotting the difference between the D14C values of MOAM and POM for all sites might be beneficial in model comparison. This figure is making it seem like the MIMICS model might be getting the bulk D14C right for the wrong reasons, but it is only one site. You could add this metric to Figure 5 and add more discussion of this difference.*
Figure 5 and Table 2 already show that MIMICS generally overestimates the D14C of POM and underestimates the D14C of MAOM while correctly predicting bulk D14C, so we think that adding the difference between the D14C values of MAOM and POM as a metric would not really add much new information. Lines 182-185 discuss how MIMICS does not accurately reproduce the D14C of POM and MAOM, but that these biases "happen to cancel out in such a way that MIMICS produces very good predictions for the D14C of bulk SOC".

*505: Details are needed on how much this difference in the original MIMICS D14C implementation affects bulk, POM and MAOM D14C values in the new implementation. Something similar to line 491 for the SOMic model should be fine.*
We added a short paragraph explaining that the error induced by the incorrect implementation is most significant during the bomb spike period and that Figure D2 clearly exhibits the effects of the incorrect implementation: the 14C curves of the litter pools are too attenuated, and MAOM had integrated more bomb-14C than POM in the 1970s (which is highly improbable). It is difficult to give actual numbers to quantify the effects of the incorrect implementation on bulk, POM and MAOM D14C (like we did for SOMic) because the effects are more erratic in this case.